# Profiling variable-number tandem repeat variation across populations using repeat-pangenome graphs

Tsung-Yu Lu [1], The Human Genome Structural Variation Consortium* & Mark J. P. Chaisson [1]✉

Variable number tandem repeats (VNTRs) are composed of consecutive repetitive DNA with hypervariable repeat count and composition. They include protein coding sequences and associations with clinical disorders. It has been difficult to incorporate VNTR analysis in disease studies that use short-read sequencing because the traditional approach of mapping to the human reference is less effective for repetitive and divergent sequences. In this work, we solve VNTR mapping for short reads with a repeat-pangenome graph (RPGG), a data structure that encodes both the population diversity and repeat structure of VNTR loci from multiple haplotype-resolved assemblies. We develop software to build a RPGG, and use the RPGG to estimate VNTR composition with short reads. We use this to discover VNTRs with length stratified by continental population, and expression quantitative trait loci, indicating that RPGG analysis of VNTRs will be critical for future studies of diversity and disease.

[1] Department of Quantitative and Computational Biology, University of Southern California, Los Angeles, CA, USA. *A list of authors and their affiliations appears at the end of the paper. ✉email: mchaisso@usc.edu

The human genome is composed of roughly 3% simple sequence repeats (SSRs)[1], loci composed of short, tandemly repeated motifs. These sequences are classified by motif length into short tandem repeats (STRs) with a motif length of six nucleotides or fewer, and variable-number tandem repeats (VNTRs) for repeats of longer motifs. SSRs are prone to hypermutability through motif copy number changes due to polymerase slippage during DNA replication[2]. Variation in SSRs are associated with tandem repeat disorders including amyotrophic lateral sclerosis and Huntington's disease[3], and VNTRs are associated with a wide spectrum of complex traits and diseases including attention-deficit disorder, Type 1 Diabetes and schizophrenia[4]. While STR variation has been profiled in human populations[5] and to find expression quantitative trait loci (eQTL)[6,7], and variation at VNTR sequences may be detected for targeted loci[8,9], the landscape of VNTR variation in populations and effects on human phenotypes are not yet examined genome-wide. Large scale sequencing studies including the 1000 Genomes Project[10], TOPMed[11] and DNA sequencing by the Genotype-Tissue Expression (GTEx) project[12] rely on short-read sequencing (SRS) characterized by SRS reads up to 150 bases. Alignment and standard approaches for detecting single-nucleotide variant (SNV) and indel variation (insertions and deletions <50 bases) using SRS are unreliable in SSR loci[13], and the majority of VNTR SVs are missed using general SV detection algorithms with SRS[14].

A number of tools have been developed specifically to detect or genotype tandem repeat variation with short reads. Most existing tools, however, support a limited description of the complexity of tandem repeats using a single motif, such as in GangSTR[15] and adVNTR[8], leaving the variation in motif sequences unexplored. While ExpansionHunter[9] allows the repeat structure to be defined by a regular expression, it is mostly restricted to STR genotyping and has not been extended to VNTRs. The full extent to which VNTR loci differ has been made more clear by long-read sequencing (LRS) and assembly. LRS assemblies have megabase scale contiguity and accurate consensus sequences[16,17] that may be used to detect VNTR variation. Nearly 70% of insertions and deletions discovered by LRS assemblies greater than 50 bases are in STR and VNTR loci[14], accounting for up to 4 Mbp per genome. Furthermore, LRS assemblies reveal how VNTR sequences differ by kilobases in length and by motif composition[18]. LRS assemblies have been used to improve VNTR analysis with SRS when used as population-specific references that add sequences missing from the reference and improve alignments[19,20]. Additionally, VNTR variation discovered by LRS assemblies may be genotyped using SRS, although with lower accuracy than other SVs[21,22]. Furthermore, the genotype represents the presence of a known variant, and does not reveal the spectrum of copy number variation that exists in tandem repeat sequences[23]. Repeat length estimation in tools specialized for tandem repeat genotyping allows more biological meaningful analyses[7,24,25].

The hypervariability of VNTRs prevents a single assembly from serving as an optimal reference. Instead, to improve both alignment and genotyping, multiple assemblies may be combined into a pangenome graph[21,23,26,27] composed of sequence-labeled vertices connected by edges such that haplotypes correspond to paths in the graph. Sequences shared between haplotypes are stored in the same vertex, and genetic variation is represented by the structure of the graph. A conceptually similar construct is the repeat graph[28], with sequences repeated multiple times in a genome represented by the same vertex. Graph analysis has been used to encode the elementary duplication structure of a genome[29] and for multiple-sequence alignment of repetitive sequences with shuffled domains[30], making them well-suited to represent VNTRs that differ in both repeat count and composition.

Here, we propose the representation of human VNTRs as a repeat-pangenome graph (RPGG), that encodes both the repeat structure and sequence diversity of VNTR loci. The most straightforward approach that combines a pangenome graph and a repeat graph is a de Bruijn graph, and was the basis of one of the earliest representations of a pangenome by the Cortex method[31,32]. The de Bruijn graph has a vertex for every distinct sequence of length $k$ in a genome ($k$-mer), and an edge connecting every two consecutive $k$-mers, thus $k$-mers occurring in multiple genomes or in multiple times in the same genome are stored by the same vertex. While the Cortex method stores entire genomes in a de Bruijn graph, we construct a separate locus-RPGG for each VNTR and store a genome as the collection of locus-RPGGs, which deviates from the definition of a de Bruijn graph because the same $k$-mer may be stored in multiple vertices.

We developed a toolkit, Tandem Repeat Genotyping based on Haplotype-derived Pangenome Graphs (danbing-tk) to identify VNTR boundaries in assemblies, construct RPGGs, align SRS reads to the RPGG, and infer VNTR motif composition and length in SRS samples. We generate a RPGG from 19 haplotype-resolved LRS genomes sequenced for population references and diversity panels[14,20,22,23], showing that while ~85% of the composition of repeats is discovered after three genomes, the genetic diversity stored in the RPGG sequentially increases as all 19 genomes are included in the RPGG. Alignment to the RPGG improves the mean absolute percentage error 28–63% over mapping to the standard human reference as a linear sequence or a repeat graph. This enables the alignment of SRS datasets into an RPGG to discover population genetics of VNTR loci, and to associate expression with VNTR variation. We find 785 loci that demonstrate population structure with respect to the inferred lengths of VNTR sequences, and importantly to discover 8,216 loci that show differential motif usage between populations. Finally, we apply danbing-tk to the SRS genomes from the GTEx consortium to discover 346 eQTL where the VNTR length is associated with gene expression.

## Results

**Repeat pan-genome graph construction.** Our approach to build RPGGs is to de novo assemble LRS genomes, and build de Bruijn graphs on the assembled sequences at VNTR loci, using SRS genomes to ensure graph quality. We used public LRS data for 19 individuals with diverse genetic backgrounds, including genomes from individual genome projects[33,34], structural variation studies[14], and diversity panel sequencing[22] (Fig. 1a and Supplementary Data 1). Each genome was sequenced by either PacBio contiguous long read, or high-fidelity sequencing between 16 and 76-fold coverage along with matched 22–82-fold Illumina sequencing (Table 1). This data reflects a wide range of technology revisions, sequencing depth, and data type, however subsequent steps were taken to ensure accuracy of RPGG through locus redundancy and SRS alignments. We developed a pipeline that partitions LRS reads by haplotype based on phased heterozygous SNVs and assembles haplotypes separately by chromosome. When available, we used existing telomere-to-telomere SNV and phase data provided by Strand-Seq and/or 10x Genomics[14,35] with phase-block N50 size between 13.4–18.8 Mb. For other datasets, long-read data were used to phase SNVs. While this data has lower phase-block N50 (<0.5–6 Mb), the individual locus-RPGG do not use long-range haplotype information and are not affected by phasing switch error. Reads from each chromosome and haplotype were independently assembled using the Flye assembler[36] for a diploid of 0.88–14.5 Mb N50, with the range of assembly contiguity reflected by the diversity of input data.

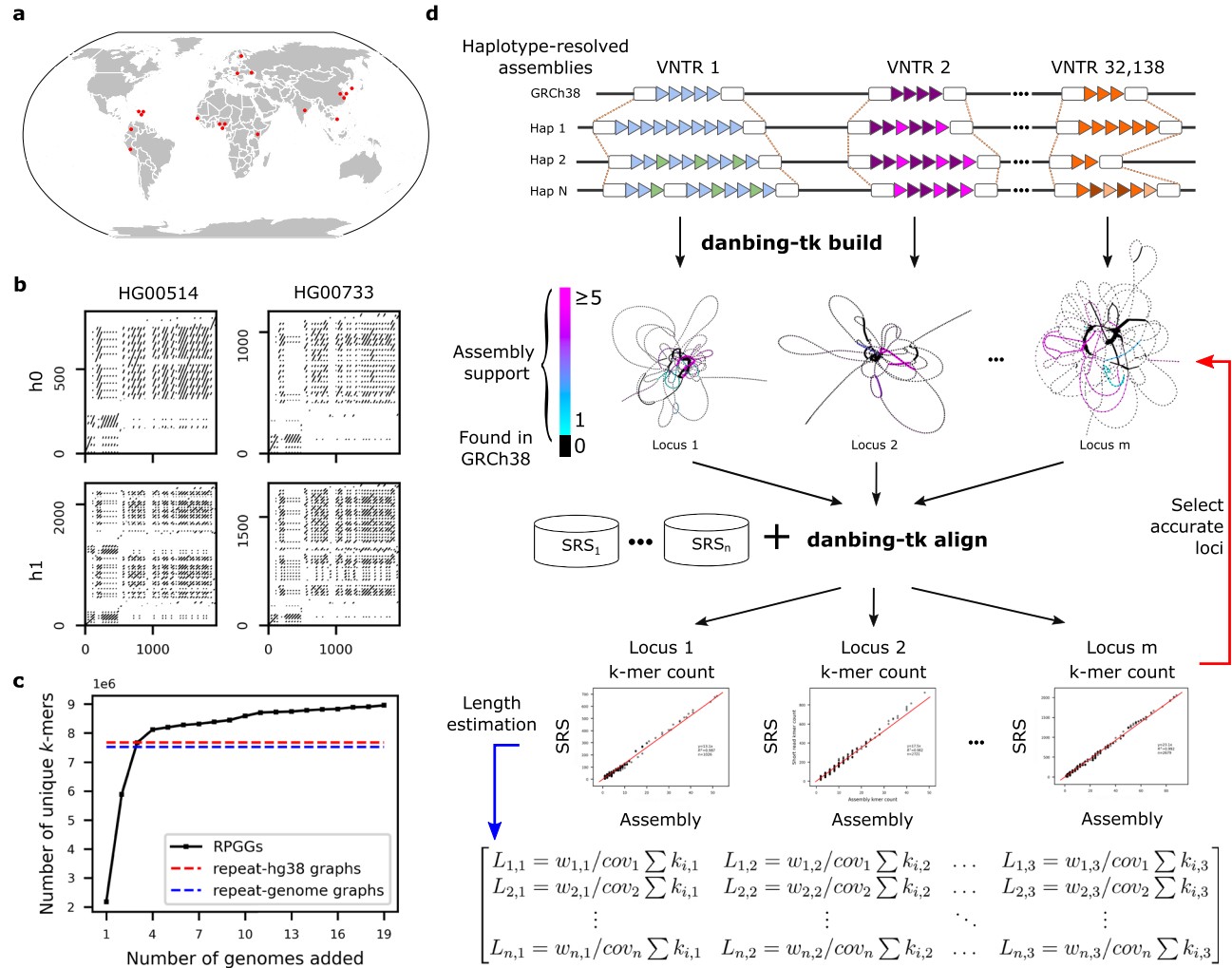

**Fig. 1 Sequence diversity of VNTRs in human populations. a** Global diversity of long-read assemblies. **b** Dot-plot analysis of the VNTR locus chr1:2280569–2282538 (SKI intron 1 VNTR) in genomes that demonstrate varying motif usage and length. **c** Diversity of RPGG as genomes are incorporated, measured by the number of *k*-mers in the 32,138 VNTR graphs. Total graph size built from GRCh38 and an average genome are also shown. **d** Danbing-tk workflow analysis. (top) VNTR loci defined from the reference are used to map haplotype loci. Each locus is converted to a de Bruijn graph, from which the collection of graphs is the RPGG. The de Bruijn graphs shown illustrate sequences missing from the RPGG built only on GRCh38. The alignments may be either used to select which loci may be accurately mapped in the RPGG using SRS that match the assemblies (red), or may be used to estimate lengths on sample datasets (blue). Source data are provided as a Source Data file.

In this study, the number of resolved VNTR loci is a more useful measurement of assembly contiguity than N50 because a disjoint RPGG is generated for each VNTR locus. An initial set of 84,411 VNTR intervals with motif size >6 bp, minimal length >150 bp and <10 kbp (mean length = 420 bp in GRCh38, Methods, Supplementary Table 1) were annotated by Tandem repeats finder (TRF)[37], and then mapped onto contig coordinates using pairwise contig alignments. This filtering criterion corresponds to an empirical cutoff of 56% purity and can retain VNTRs (*n* = 2715, Supplementary Fig. 1) that have nested STR annotations (Supplementary Fig. 2). Long VNTR loci tended to have fragmented TRF annotation, which can cause erroneous length estimates in downstream analysis and fail to properly interpret repeat structures as a whole such as in adVNTR-NN (Supplementary Fig. 3). During locus assignment, danbing-tk expands boundaries and merges loci to ensure boundaries of all VNTRs are well-defined and harmonized across genomes (Methods) (Fig. 1b). In practice, we found that 43,869/84,411 (52%) of the VNTR loci are subject to boundary expansion, with an average expansion size of 539 bp. The set of VNTRs that can

be properly annotated ranges from 19,800 to 73,212 depending on the assembly quality, with a final set of 73,582 loci (mean length = 652 bp) across 19 genomes (Supplementary Fig. 4 and Supplementary Data 2).

The RPGGs are a collection of independently constructed bidirectional de Bruijn graphs of each VNTR locus and flanking 700 bases from the haplotype-resolved assemblies. In a bidirectional de Bruijn graph, each distinct sequence of length *k* (*k*-mer) and its reverse complement map to a vertex, and each sequence of length *k* + 1 connects the vertices to which the two composite *k*-mers map. The RPGG differs from a standard bidirectional de Bruijn graph because a *k*-mer may be repeated in multiple subgraphs. There was little effect on downstream analysis for values of *k* between 17 and 25, and so *k* = 21 was used for all applications. To remove spurious vertices and edges from assembly consensus errors, SRS from genomes matching the LRS samples were mapped to the RPGG, and *k*-mers not mapped by SRS were removed from the graph (average of 264 per locus). Using the number of vertices as a proxy for sampled genetic diversity, we find that 27%

**Table 1 Source genomes for the RPGG.**

| Genome | Continental population | Study | Coverage | Assembly N50 (Mb) | Fraction of VNTR annotated | Ancestry |
|---|---|---|---|---|---|---|
| AK1 | EAS | KG | 54 | 0.88 | 0.840 | Korean |
| HG00268 | EUR | DP | 67 | 3.51 | 0.967 | Finnish |
| HG00512 | EAS | HGSVG | 28 | 8.83 | 0.995 | Han Chinese |
| HG00513 | EAS | HGSVG | 30 | 1.57 | 0.993 | Han Chinese |
| HG00514 | EAS | HGSVG | 31 | 1.32 | 0.948 | Han Chinese |
| HG00731 | AMR | HGSVG | 31 | 2.18 | 0.995 | Puerto Rican |
| HG00732 | AMR | HGSVG | 16 | 1.3 | 0.992 | Puerto Rican |
| HG00733 | AMR | HGSVG | 46 | 6.88 | 0.992 | Puerto Rican |
| HG01352 | AMR | DP | 68 | 5.97 | 0.992 | Colombian |
| HG02059 | EAS | DP | 76 | 19.5 | 0.992 | Vietnamese |
| HG02106 | AMR | DP | 57 | 0.88 | 0.640 | Peruvian |
| HG02818 | AFR | DP | 56 | 0.66 | 0.802 | Gambian |
| HG04217 | SAS | DP | 60 | 0.86 | 0.269 | Telugu |
| NA12878 | EUR | DP | 54 | 4.67 | 0.971 | Central European |
| NA19238 | AFR | HGSVG | 23 | 2.64 | 0.991 | Yoruba |
| NA19239 | AFR | HGSVG | 35 | 4.87 | 0.994 | Yoruba |
| NA19240 | AFR | HGSVG | 49 | 3.4 | 0.989 | Yoruba |
| NA19434 | AFR | DP | 62 | 11 | 0.980 | Luhya |
| NA24385 | EUR | GIAB | 54 | 1.32 | 0.981 | Ashkenazim |

Continental populations represented are East Asian (EAS), European (EUR), Admixed Amerindian (AMR), South Asian (SAS), and African (AFR). Coverage is estimated diploid coverage based on alignment to GRCh38. Assembly N50 is of haplotype-resolved assemblies. The fraction of VNTR annotated are all VNTR with at least 700 flanking bases assembled.

(2,102,270 new nodes) of the sequences not contained in GRCh38 (7,672,357 nodes) are discovered after the inclusion of 19 genomes, with diversity linearly increasing per genome after the first four genomes are added to the RPGG (8,958,361 nodes, Fig. 1c).

The alignment of a read to an RPGG may be defined by the path in the RPGG with a sequence label that has the minimum edit distance to the read among all possible paths. We used $\sim 5.88 \times 10^8$ error-free 150 bp paired-end reads simulated from six genomes (HG00512, HG00513, HG00731, HG00732, NA19238, and NA19239) to evaluate how reads are aligned to the RPGG. While methods exist to find alignments that do not reuse cycles[38], others allow alignment to cyclic graphs but with high computational costs when applied to RPGG[27] or are limited to alignment in STR regions[9]. Efficient alignment with cycles is a more challenging problem recently solved by GraphAligner[39] to map long reads to pangenome graphs. Although >99.99% of the reads simulated from VNTR loci were aligned, 6.03% of reads matched with less than 90% identity, indicating misalignment. We developed an alternative approach tuned for RPGG alignments in danbing-tk (Fig. 1d) to realign all SRS reads within a bam/fastq file to the RPGG in two passes, first by finding locus-RPGGs with a high number (>45 in each end) shared $k$-mers with reads, and next by threading the paired-end reads through the locus-RPGG, allowing for up to two edits (mismatch, insertion, or deletion) and at least 50 matched k-mers per read against the threaded path (Methods). Using danbing-tk, 99.997% of VNTR-simulated reads were aligned with >90% identity. The RPGG is only built on VNTRs and their flanking sequences, excluding the rest of the genome. When reads from the entire genome are considered, for 96.6% of the loci (71,080/73,582), danbing-tk can map >90% of the reads back to their original VNTR regions. Misaligned reads from either other VNTR loci included in the RPGG or the remainder of the genome not included in the RPGG target relatively few loci; 3.6% (2,635/73,582) loci have at least one read misaligned from outside the locus. The graph pruning step is the primary cause of missed alignments, and affects on average 2,772 loci per assembly. On real data, danbing-tk required 18.5 GB of memory to map 150 base paired-end reads at 10.1 Mb/sec on 16 cores.

**Read-to-graph alignment in VNTR regions**. Alignment of SRS reads to the RPGG enables estimation of VNTR length and motif composition. The count of $k$-mers in SRS reads mapped to the RPGG are reported by danbing-tk for each locus. For $M$ samples and $L$ VNTR loci, the result of an alignment is $L$ count matrices of dimension $M \times N_i$, where $N_i$ is the number of vertices in the de Bruijn graph on the locus $i$, excluding flanking sequences. If SRS reads from a genome were sequenced without bias, sampled uniformly, and mapped without error to the RPGG, the count of a $k$-mer in a locus mapped by an SRS sample should scale by a factor of read depth with the sum of the count of the $k$-mer from the locus of both assembled haplotypes for the same genome. The quality of alignment (aln-$r^2$) and sequencing bias were measured by comparing the $k$-mer counts from the 19 matched Illumina and LRS genomes (Fig. 2a). In total, 44% (32,138/73,582) loci had a mean aln-$r^2 \geq 0.96$ between SRS and assembly $k$-mer counts, and were marked as "valid" loci to carry forward for downstream diversity and expression analysis (Fig. 2b). Valid had an average length of 341 bp, compared to 657 bp in the entire database (Fig. 2c). VNTR loci that did not align well (invalid) were enriched for sequences that map within Alu (21,820), SVA (1762), and other 26,752 mobile elements (Supplementary Fig. 3); loci with false mapping in the simulation experiment are also enriched in the invalid set (Supplementary Table 2). Specifically, 71.6% (4,297/5,999) of loci with false-positive mapping, 84.7% (8,065/9,525) of loci with false-negative mapping are not marked as valid. Loci with false mapping but retained in the final set have lower but still decent length-prediction accuracy (0.79 versus 0.82). The complete RPGG on valid loci contains 8,958,361 vertices, in contrast to the corresponding RPGG on GRCh38 only (repeat-GRCh38), which has 7,672,357 vertices. We validate that the additional vertices in the RPGG are indeed important for accurately recruiting reads pertinent to a VNTR locus, using the *CACNA1C* VNTR as an example (Fig. 2d). It is known that the reference sequence at this locus is truncated compared to the majority of the populations (319 bp in GRCh38 versus 5669 bp averaged across 19 genomes). The limited sequence diversity provided by repeat-GRCh38 at this locus failed to recruit reads that map to paths existing in the RPGG but missing or only partially represented in repeat-GRCh38. A linear fit between the

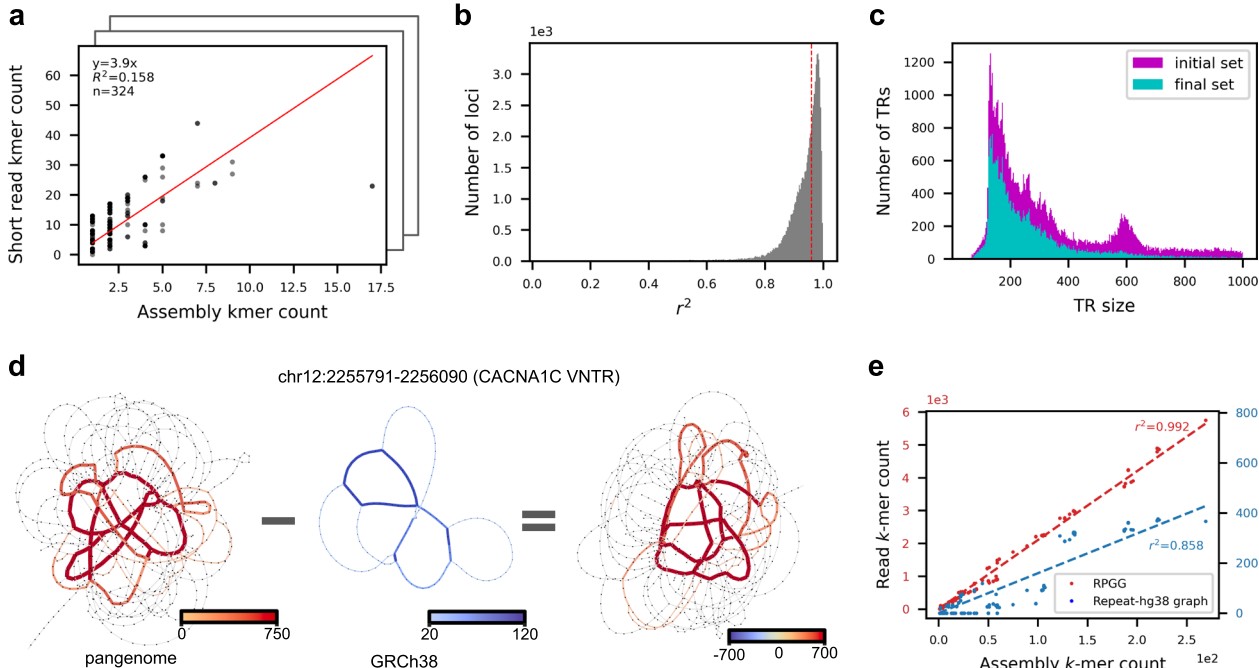

**Fig. 2 Mapping short reads to repeat-pangenome graphs. a** An example of evaluating the alignment quality of a locus mapped by SRS reads. The alignment quality is measured by the $r^2$ of a linear fit between the $k$-mer counts from the ground truth assemblies and from the mapped reads (Methods). **b** Distribution of the alignment quality scores of 73,582 loci. Loci with alignment quality less than 0.96 when averaged across samples are removed from downstream analysis (Methods). **c** Distribution of VNTR lengths in GRCh38 removed or retained for downstream analysis. **d, e** Comparing the read mapping results of the *CACNA1C* VNTR using RPGG or repeat-GRCh38. **d** The $k$-mer counts in each graph and the differences are visualized with edge width and color saturation. To visualize paths with less mapped reads, $k$-mer counts are clipped at 750 (left), 120 (middle), and 700 (right), respectively, with the maximal $k$-mer count of each graph being 5744, 375, and 5378, respectively. **e** The $k$-mer counts from the ground truth assemblies are regressed against the counts from reads mapped to the RPGG (red) and repeat-GRCh38 (blue), respectively. Source data are provided as a Source Data file.

$k$-mers from mapped reads and the ground truth assemblies shows that there is a 13-fold gain in slope, or measured read depth, when using RPGG compared to repeat-GRCh38 (Fig. 2e). The $k$-mer counts in the RPGGs also correlate better with the assembly $k$-mer counts compared to the repeat-GRCh38 (aln-$r^2 = 0.992$ versus 0.858).

New genomes with arbitrary combinations of motifs and copy numbers in VNTRs should still align to an RPGG as long as the motifs are represented in the graph. We used leave-one-out analysis to evaluate alignment of unseen genomes to RPGGs and estimation of VNTR length. In each experiment, an RPGG was constructed with one LRS genome missing. SRS reads from the missing genome were mapped into the RPGG, and the estimated locus lengths were compared to the average diploid lengths of corresponding loci in the missing LRS assembly. The locus length is estimated as the adjusted sum of $k$-mer counts kms mapped from SRS sample $s$: $\text{kms}/(\text{cov}_s \times \hat{b})$, where $\text{cov}_s$ is sequencing depth of $s$, $\hat{b}$ is a correction for locus-specific sampling bias (LSB). LSB measures the deviation of an observed read depth from the expected value within an interval (see "Methods" for formal definition). As the SRS datasets used in this study during pangenome construction were collected from a wide variety of studies with different biases, there was no consistent LSB in either repetitive or nonrepetitive regions for samples from different sequencing runs (Supplementary Figs. 6 and 7). However, principal component analysis (PCA) of repetitive and nonrepetitive regions showed highly similar projection patterns (Supplementary Fig. 8), which enabled using LSB in nonrepetitive regions as a proxy for finding the nearest neighbor of LSB in VNTR regions (Supplementary Fig. 9). Leveraging this finding, a set of 397 nonrepetitive control regions were used to estimate the LSB

of an unseen SRS sample (Methods), giving a median length-prediction accuracy of 0.82 for 16 unrelated genomes (Fig. 3a left, Supplementary Fig. 10). The read depth of a repetitive region correlates to the locus length when aligning short reads to a linear reference genome. However, estimation of VNTR length from read depth has an accuracy of 0.75 (Fig. 3a left). We also compared the performance for length prediction using the RPGG versus repeat-GRCh38, and observed a 58% improvement in accuracy (0.82 versus 0.52, Fig. 3a left and Supplementary Fig. 11). The overall error rate, measured with mean absolute percentage error (MAPE), of all loci ($n = 32,138$) are also significantly lower when using RPGGs (MAPE = 0.18, Fig. 3a right) compared with the repeat-GRCh38 (0.23, paired *t*-test $P = 4.2 \times 10^{-32}$) or reference-aligned read depth (0.20, paired *t*-test $P = 2.4 \times 10^{-33}$). Furthermore, a 62% reduction in error size is observed for the 6,383 loci poorly genotyped (MAPE > 0.4) using repeat-GRCh38 (Fig. 3b, MAPE = 0.235 versus 0.610). Loci with low accuracy in length estimates from RPGG can be mostly explained by the estimation error in LSB due to varying data quality ($r^2 = 0.89$, Supplementary Fig. 12; example given in Supplementary Fig. 13), and to a slight degree by the presence of a missing haplotype (Supplementary Fig. 14), the fraction of $k$-mers in a locus unique to a sample (Supplementary Fig. 15), GC bias (Supplementary Fig. 16), and the difference in the VNTR GC content across samples (Supplementary Fig. 17).

**Profiling VNTR length and motif diversity.** To explore global diversity of VNTR sequences and potential functional impact, we aligned reads from 2,504 individuals from diverse populations sequenced at 30-fold coverage from the 1000-Genomes project (1KGP)[10,40], and 879 GTEx genomes[12] to the RPGG. The

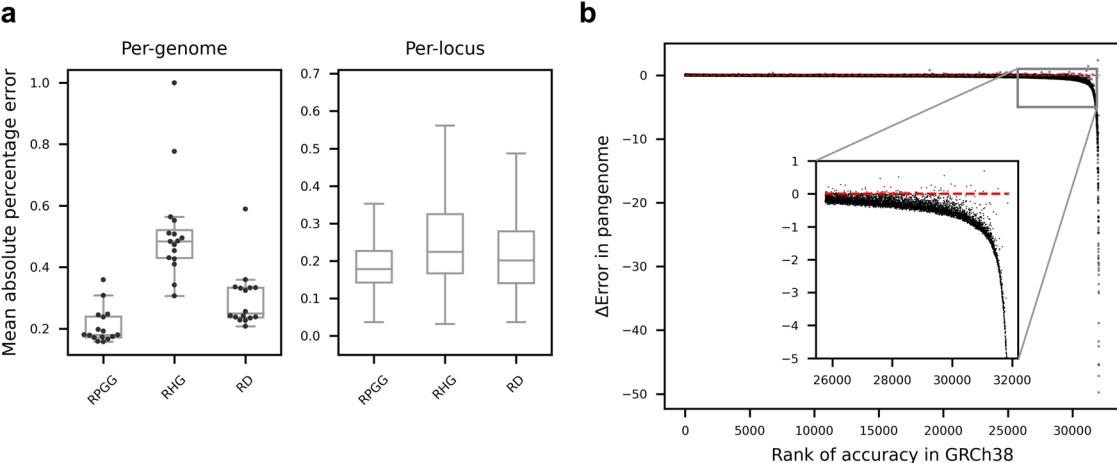

**Fig. 3 VNTR length prediction. a** Accuracies of VNTR length-prediction measured for each genome (left; $n = 16$) and each locus (right; $n = 32,138$). Mean absolute percentage error (MAPE) in VNTR length is averaged across loci and genomes, respectively. Lengths were predicted based on repeat-pangenome graphs (RPGG), repeat-GRCh38 (RHG) or naive read depth method (RD), respectively. Boxes span from the lower quartile to the upper quartile, with horizontal lines indicating the median. Whiskers extend to points that are within 1.5 interquartile range (IQR) from the upper or the lower quartiles. **b** Relative performance of RPGG versus repeat-GRCh38. Loci are ordered along the x-axis by genotyping accuracy in repeat-GRCh38. The y-axis shows the decrease in MAPE using RPGG versus repeat-GRCh38. The subplot shows loci poorly genotyped (MAPE > 0.4) in repeat-GRCh38. The red dotted line indicates the baseline without any improvement. the counts from reads mapped to the RPGG (red) and repeat-GRCh38 (blue), respectively. Source data are provided as a Source Data file.

fraction of reads from these datasets that align to the RPGG ranges from 1.11 to 1.37%, similar to the matched LRS/SRS data (1.23%). PCA on the LSB of both datasets showed the 1KGP and GTEx genomes as separate clusters in both repetitive and non-repetitive regions (Supplementary Fig. 7), indicating experiment-specific bias that prevents cross dataset comparisons. Consistent with the finding in previous leave-one-out analysis, genomes from the same study cluster together in the PCA plot of LSB, and so within each dataset and locus, k-mer counts from SRS reads normalized by sequencing depth were used to compare VNTR content across genomes.

The k-mer dosage: kms/cov, was used as a proxy for locus length to compare tandem repeat variation across populations in the 1KGP genomes. The 1KGP samples contain individuals from African (26.5%), East Asian (20.1%), European (19.9%), Admixed American (13.9%), and South Asian (19.5%) populations. When comparing the average population length to the global average length, 60.8% (19,530/32,138) have differential length between populations (FDR = 0.05 on ANOVA P-values), with similar distributions of differential length when loci are stratified by the accuracy of length prediction (Fig. 4a). Population stratification was calculated using the $V_{ST}$ statistic[41] on VNTR length (Fig. 4b). Previous studies have used >3 standard deviations above the mean to define for highly stratified copy number variants[42]. Under this measure, 785 variants are highly stratified, including 266 that overlap genes, however this is not significantly enriched ($P = 0.079$, one-sided permutation test). Two of the top five loci ranked by $V_{ST}$ are intronic: a 72 base VNTR in *PLCL1* ($V_{ST} = 0.37$), and a 148 base locus in *SPATA18* ($V_{ST} = 0.35$) (Fig. 4c, d). These values for $V_{ST}$ are lower than what are observed for large copy number variants[41] and may be the result of neutral variation, however, this may be affected by the high variance of the length estimate, as $V_{ST}$ decreases as the variance of the copy number/dosage values increase (Methods).

VNTR loci that are unstable may undergo hyper-expansion and are implicated as a mechanism of multiple diseases[4]. To discover potentially unstable loci, we searched the 1KGP genomes for evidence of rare VNTR hyper-expansion. Loci were screened for individuals with extreme (>6 standard deviations)

variation, and then filtered for deletions or unreliable samples (Methods) to characterize 477 loci as potentially unstable (Supplementary Data 3). These loci are inside 115 genes and are significantly reduced from the number expected by chance ($P < 1 \times 10^{-5}$, one-sided permutation test; $n = 10,000$). Of these loci, 64 have an individual with >10 standard deviations above the mean, of which two overlap genes, *KCNA2*, and *GRM4* (Supplemental Fig. 18).

Alignment to an RPGG provides information about motif usage in addition to estimates of VNTR length because genomes with different motif composition will align to different vertices in the graph. To detect differential motif usage, we searched for loci with a k-mer that was more frequent in one population than another and not simply explained by a difference in locus length, comparing African (AFR) and East Asian populations for maximal genetic diversity. Lasso regression against locus length was used to find the k-mer with the most variance explained (VEX) in EAS genomes, denoted as the most informative k-mer (mi-kmer). Two statistics are of interest when comparing the two populations: the difference in the count of mi-kmers ($kmc_d$) and the difference between proportion of VEX ($r_d^2$) by mi-kmers. $Kmc_d$ describes the usage of an mi-kmer in one population relative to another, while $r_d^2$ indicates the degree that the mi-kmer is involved in repeat contraction or expansion in one population relative to another. We observe that 8216 loci have significant differences in the usage of mi-kmers between the two populations (two-sided $P < 0.01$, bootstrap, Supplementary Fig. 19). Among these, the mi-kmers of 1913 loci are crucial to length variation in the EAS but not in the AFR population (two-sided $P < 0.01$, bootstrap) (Fig. 4e and Supplementary Fig. 19). A top example of these loci with $r^2$ at least 0.9 in the EAS population was visualized with a heatmap of relative k-mer count from both populations, and clearly showed differential usage of cycles in the RPGG (Fig. 4f).

**Association of VNTR with nearby gene expression.** As the danbing-tk length estimates showed population genetic patterns expected for human diversity, we assessed whether danbing-tk

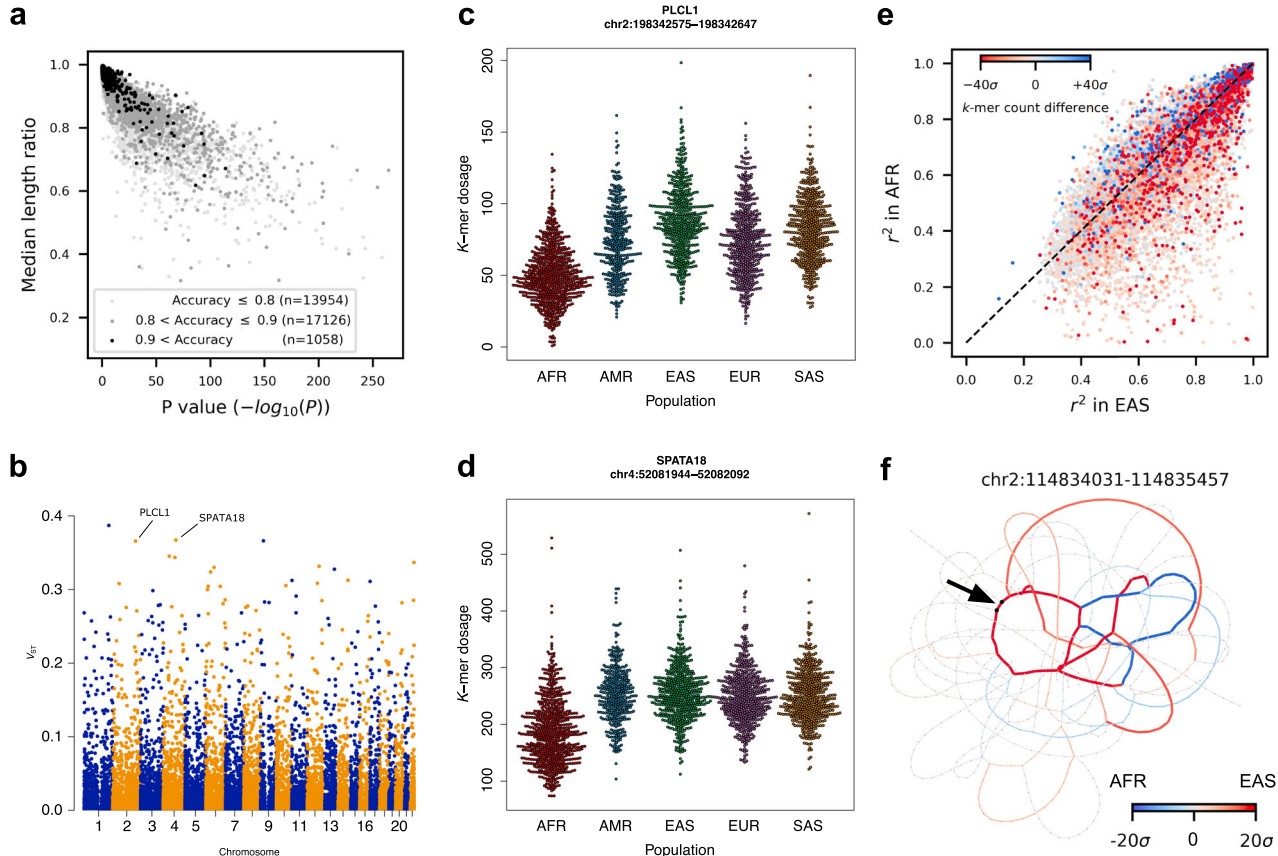

**Fig. 4 Population properties of VNTR loci. a** Ratios of median length between populations for loci with significant differences in average length. Loci are stratified by accuracy prediction (<0.8), medium (0.8–0.9), and high (0.9+). **b** Manhattan plot of $V_{ST}$ values. **c, d** The distribution of estimated length via $k$-mer dosage in continental populations for *PLCL1* and *SPATA18* VNTR loci, selected to visualize the distribution of dosage in different populations. Each point is an individual. **e** Differential usage and expansion of motifs between the EAS and AFR populations. For each locus, the proportion of variance explained by the most informative $k$-mer in the EAS is shown for the EAS and AFR populations on the x- and y-axes, respectively. Points are colored by the difference in normalized $k$-mer counts, with red and blue indicating $k$-mers more abundant in EAS and AFR populations, respectively. **f** An example VNTR with differential motif usage. Edges are colored if the $k$-mer count is biased toward a certain population. The black arrow indicates the location of the $k$-mer that explains the most variance of VNTR length in the EAS population. Source data are provided as a Source Data file.

alignments could detect VNTR variation with functional impact. Genomes from the GTEx project were mapped into the RPGG to discover loci that have an effect on nearby gene expression in a length-dependent manner. A total of 812/838 genomes with matching expression data passed quality filtering (Methods). Similar to the population analysis, the $k$-mer dosage was used as a proxy for locus length. Methods previously used to discover eQTL using STR genotyping[6] were applied to the danbing-tk alignments. In sum, 30,362 VNTRs within 100 kb to 45,720 GTEx gene-annotations (including genes, lncRNA, and other transcripts) were tested for association, with a total of 149,057 tests and ~3.3 VNTRs tested per gene. Using a gene-level FDR cutoff of 5%, we find 346 eQTL (eVNTRs) (Fig. 5a), among which 344 (99.4%) discoveries are previously unreported (Supplementary Table 3), indicating that the spectrum of association between tandem repeat variation and expression extends beyond the lengths and the types of SSR considered in previous STR[15] and VNTR[43] studies. This likely represents a lower bound on the total eVNTRs as analysis on the complete set of loci that are not filtered by mapping accuracy include 658 eVNTRs (Supplementary Data 4–6). Both positive and negative effects were observed among eVNTRs (Fig. 5b). More eVNTRs with positive effect size were found than with a negative effect size (200 versus 146, binomial test $P = 0.0043$), with an average effect of $+0.261$ (from $+0.139$ to $+0.720$) versus $-0.247$ (from $-0.524$ to $-0.159$),

respectively. eVNTRs tend to be closer to telomeres relative to all VNTRs (Mann–Whitney $U$-test $P = 5.2 \times 10^{-5}$, Supplementary Fig. 20). As many exons contain VNTR sequences, expression measured by read depth should increase with length of the VNTR, and there is an 2.5-fold enrichment of eVNTRs in coding regions as expected.

The eVNTRs have the potential to yield insight to disease. In one example, an intronic eVNTR at chr5:96,896,863–96,896,963 flanks exon 9 of *ERAP2* (Fig. 5d and Supplementary Fig. 21). The eVNTR has a $-0.52$ effect size and was reported across 27 tissues. Although the effect is not independent of the lead eSNP (Supplementary Figs. 22 and 23), the variant is missing from the GTEx *cis*-eQTL catalog and colocalizes with a regulatory hotspot with peaks of histone markers, DNase and 40 different ChIP signals. The protein product of *ERAP2*, or endoplasmic reticulum aminopeptidase 2, is a zinc metalloaminopeptidase involving in the process of Class I MHC mediated antigen presentation and innate immune response. It has been reported to be associated with several diseases including ankylosing spondylitis[44] and Crohn's disease[45]. Abnormal expansion of the VNTR might increase autoimmune disease risk through reducing *ERAP2* expression, leaving longer and more antigenic peptides, yet potentially higher fitness against virus infection[46]. This VNTR is a unique sequence in GRCh38 that is a 101 bp tandem duplication in 17/38 of the haplotypes. Another example is an

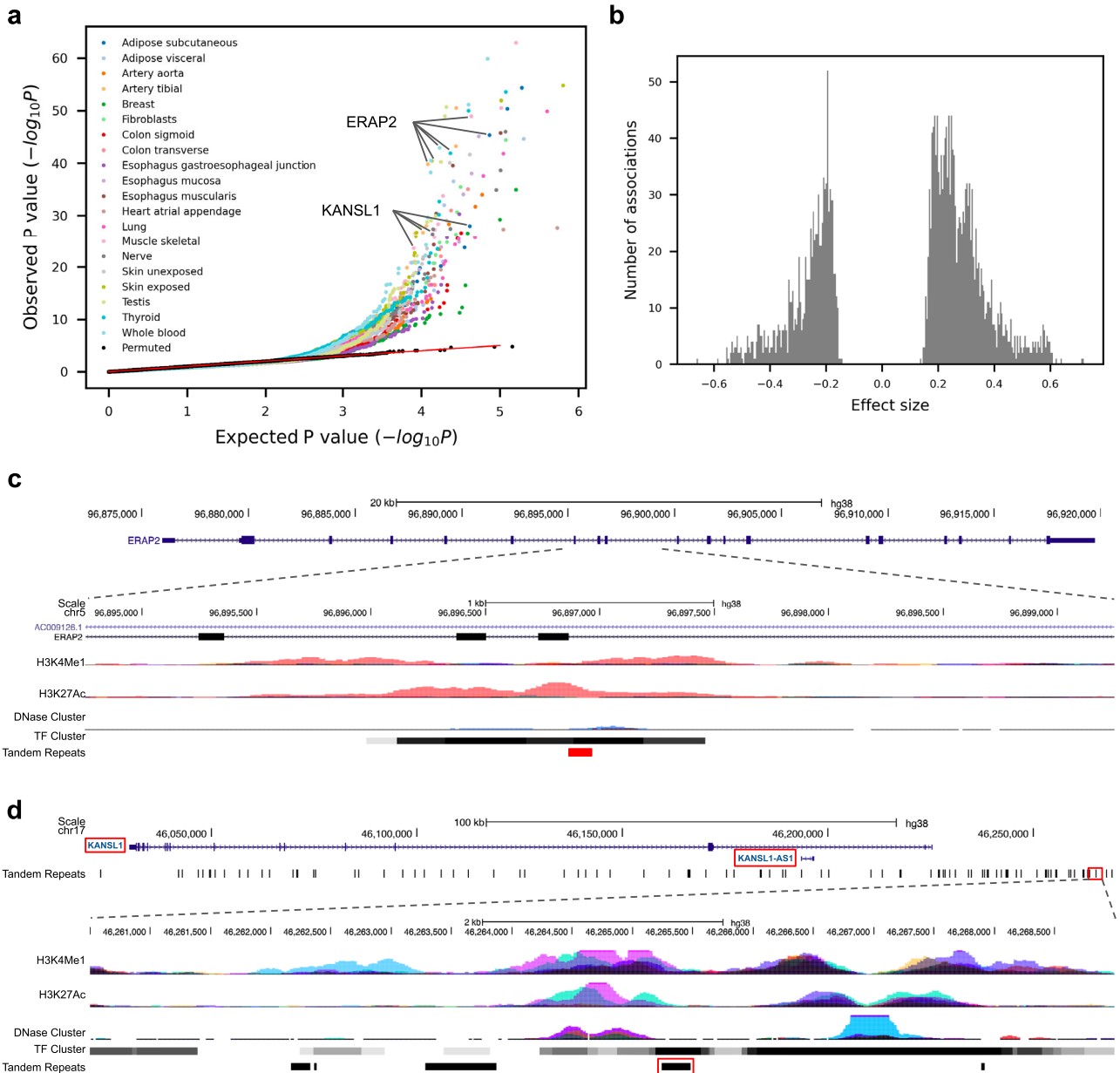

**Fig. 5 *cis*-eQTL mapping of VNTRs. a** eVNTR discoveries in 20 human tissues. The quantile-quantile plot shows the observed *P*-value of each association test (two-sided *t*-test) versus the *P*-value drawn from the expected uniform distribution. Black dots indicate the permutation results from the top 5% associated (gene, VNTR) pairs in each tissue. The regression plots for *ERAP2* and *KANSL1* are shown in **c** and **d**. **b** Effect size distribution (*n* = 2510) of significant associations from all tissues. **c**, **d** Genomic view of disease-related (eGene,eVNTR) pairs (*ERAP2*, chr5:96896863–96896963) (**c**) and (*KANSL1*, chr17:46265245–46265480) (**d**) are shown. Red boxes indicate the location of eGenes and eVNTRs.

intergenic VNTR at chr17:46,265,245–46,265,480 that associates with the expression of *KANSL1* ~40 kb upstream (Fig. 5c and Supplementary Fig. 21). The eVNTR has a maximal effect size of +0.45 and is significant across 40 tissues. The protein product of *KANSL1*, or KAT8 regulatory NSL complex subunit 1, is a part of the histone acetylation machinery. Deletion of this gene is linked to Koolen-de Vries syndrome[47], and the locus is associated with Parkinson disease[48]. The eVNTR colocalizes with strong ChIP signals, and the association of this VNTR with the epigenetic landscape warrants further investigation.

## Discussion
Previous commentaries have proposed that variation in VNTR loci may represent a component of undiagnosed disease and missing heritability[49], which has remained difficult to profile even

with whole-genome sequencing[15]. To address this, we have proposed an approach that combines multiple genomes into a pangenome graph that represents the repeat structure of a population. This is supported by the software, danbing-tk and associated RPGG. We used danbing-tk to generate a pangenome from 19 haplotype-resolved assemblies, and applied it to detect VNTR variation across populations and to discover eQTL.

The structure of the RPGG can help to organize the diversity of assembled VNTR sequences with respect to the standard reference. In particular, the RPGG on 19 genomes is 27% larger than repeat-GRCh38. Combined with the observation that using the 19-genome RPGG gives a 63% decrease in length-prediction error, this indicates that the pan-genomes add detail for the missing variation. With the availability of additional genomes sequenced through the Pangenome Reference Consortium

(https://humanpangenome.org/) and the HGSVC (https://www.internationalgenome.org), combined with advanced haplotype-resolve assembly methods[50], the spectrum of this variation will be revealed in the near future. While we anticipate that eventually the full spectrum of VNTR diversity will be revealed through LRS of large cohorts, the RPGG analysis will help organize analysis by characterizing repeat domains. For example, with our approach, we are able to detect 1913 loci with differential motif usage between populations, which could be difficult to characterize using an approach such as multiple-sequence alignment of VNTR sequences from assembled genomes.

There are several caveats to our approach. Datasets combined from disparate sequencing runs with batch effects will affect dosage estimates. In contrast to other pangenome approaches[27,38], danbing-tk does not keep track of a reference (e.g., GRCh38) coordinate system. Furthermore, because it is often not possible to reconstruct a unique path in an RPGG, only counts of mapped reads are reported rather than the order of traversal of the RPGG. An additional caveat of our approach is that genotype is calculated as a continuum of $k$-mer dosage rather than discrete units, prohibiting direct calculation of linkage-disequilibrium for fine-scale mapping[51]. Finally this approach only profiles loci where $k$-mer counts from reads and assemblies are correlated; loci for which every $k$-mer appears the same number of times are excluded from analysis (on average 8,058/73,582 per genome).

The rich data (Supplementary Data 7–9) provided by danbing-tk and pangenome analysis provide the basis for additional association studies. While most analysis in this study focused on the diversity of VNTR length or association of length and expression, it is possible to query differential motif usage using the RPGG. The ability to detect motifs that have differential usage between populations brings the possibility of detecting differential motif usage between cases and controls in association studies. This can help distinguish stabilizing versus fragile motifs[52], or resolve some of the problem of missing heritability by discovering new associations between motif and disease[18]. Finally, this work is a part of ongoing pangenome graph analysis[53,54], and represents an approach to generating pangenome graphs in loci that have difficult multiple-sequence alignments or degenerate graph topologies. Additional methods may be developed to harmonize danbing-tk RPGGs with genome-wide pangenome graphs constructed from other methods.

## Methods

### Repeat-pangenome graph construction

*Initial discovery of tandem repeats.* TRF[37] v4.09 (option: 2 7 7 80 10 50 500 -f -d -h) was used to roughly annotate the SSR regions of five PacBio assemblies (AK1, HG00514, HG00733, NA19240, NA24385). The scope of this work focuses on VNTRs that cannot be resolved by typical short-read sequencing methods. We selected the set of SSR loci with a motif size greater than 6 bp and a total length greater than 150 bp and <10 kbp. For each haplotype, the selected VNTR loci were mapped to GRCh38 reference genome to identify homologous VNTR loci. To maintain data quality, VNTR loci that could not be assigned homology were removed from datasets.

*Boundary expansion of VNTRs.* The biological boundaries of a VNTR are ill-defined; VNTRs with sparse recurring motifs or transition between different motifs or a nested motif structure often fail to be fully annotated by TRF. A misannotation of VNTR boundaries can cause erroneous length estimates. To avoid the propagation of this error to downstream analysis, we developed a multiple boundary expansion algorithm to recover the proper boundary for each VNTR across all haplotypes, including the 14 remaining genomes (HG00268, HG00512, HG00513, HG00731, HG00732, HG01352, HG02059, HG02106, HG02818, HG04217, NA12878, NA19238, NA19239 and NA19434). The algorithm maintains an invariant: the flanking sequence in any of the haplotypes does not share $k$-mers with the VNTR regions from all haplotypes. VNTR boundaries in each haplotype are iteratively expanded until the invariant is true or if expansion exceeds 10 kbp in either 5′ or 3′ direction. The size of the flanking regions is chosen to be 700 bp, which is approximately the upper bound of the insert size of typical SRS reads. The

following QC step removes a haplotype if its VNTR annotation is within 700 bp to breakpoints or if the orthology mapping location to GRCh38 is different from the majority of haplotypes. A VNTR locus with the number of supporting haplotypes less than 90% of the total number of haplotypes is also removed. Adjacent VNTR loci within 700 bp to each other in any of the haplotypes will induce a merging step over all haplotypes. Haplotypes with distance between adjacent loci inconsistent with the majority of haplotypes are removed. Finally, VNTR loci with the number of supporting haplotypes less than 80% of the total number of haplotypes are removed, leaving 73,582 of the initial 84,411 loci.

*Read-to-graph alignment.* For the two haplotypes of an individual, three data structures are used to encode the information of all VNTR loci, including VNTRs and their 700 bp flanking sequences. The first data structure allows fast locus lookup for each $k$-mer ($k = 21$) by hashing each canonical $k$-mer in the VNTRs and the flanking sequences to the index of the original locus. The second data structure enables graph threading by storing a bi-directional de Bruijn graph for each locus. The third data structure is used for counting $k$-mers originating from VNTRs. The read mapping algorithm maps each pair of Illumina paired-end reads to a unique VNTR locus in three phases: (1) In the $k$-mer set mapping phase, the read pair is converted to a pair of canonical $k$-mer multisets. The VNTR locus with the highest count of intersected $k$-mers is detected with the first data structure. (2) In the threading phase, the algorithm tries to map the $k$-mers in the read pair to the bi-directional de Bruijn graph such that the mapping forms a continuous path/cycle. To account for sequencing and assembly errors, the algorithm is allowed to edit a limited number of nucleotides in a read if no matching $k$-mer is found in the graph. The read pair is determined feasible to map to a VNTR locus if the number of mapped $k$-mers is above an empirical threshold. (3) In the $k$-mer counting phase, canonical $k$-mers of the feasible read pair are counted if they existed in the VNTR locus. Finally, the read mapping algorithm returns the $k$-mer counts for all loci as mapped by SRS reads. Alignment timing was conducted on an Intel Xeon E5-2650v2 2.60 GHz node.

*Graph pruning and merging.* Pan-genome representation provides a more thorough description of VNTR diversity and reduces reference allele bias, which effectively improves the quality of read mapping and downstream analysis. Considering the fact that haplotypes assembled from long-read datasets are error prone in VNTR regions, it is necessary to prune the graphs/$k$-mers before merging them as a pan-genome. We ran the read mapping algorithm with error correction disabled so as to detect $k$-mers unsupported by SRS reads. The three data structures were updated by deleting all unsupported $k$-mers for each locus. By pooling and merging the reference regions corresponding to the VNTR regions in all individuals, we obtained a set of "pan-reference" regions, each indicating a location in GRCh38 that is likely to map to a VNTR region in any other unseen haplotype. By referencing the mapping relation of VNTR loci across individuals, we encoded the variability of each VNTR locus by merging the three data structures across individuals.

*Alignment quality analysis.* To evaluate the quality of the haplotype assemblies and the performance of the read mapping algorithm, VNTR $k$-mer counts in the original assemblies were regressed against those mapped from SRS reads. The $r^2$ of the linear fit was used as the alignment quality score (referred to as aln-$r^2$). To measure alignment quality in the pan-genome setting, only the $k$-mer set derived from the genotyped individual was retained as the input for regression.

*Data filtering.* A final set of 32,138 VNTR regions was called by filtering based on aln-$r^2$. The quality of a locus was measured by the mean aln-$r^2$ across individuals. Loci with mean aln-$r^2$ below 0.96 were removed from the final call set. The final pan-genome graphs were used to genotype large Illumina datasets, measure length-prediction accuracy, analyze population structures and map eQTL.

*Predicting VNTR lengths.* Read depths at VNTR regions usually vary considerably from locus to locus. Furthermore, the sampling bias of different sequencing runs are also different, which limits our ability to genotype the accurate length of VNTRs. To account for this, we compute LSBs $b_s$ for each sample $s$, which is a tuple of (genome $g$, sequencing run), as follows:

$$b_s = \frac{\mathrm{kms}_s}{\mathrm{cov}_s \times L_g} \qquad (1)$$

where $L_g$ is the ground truth VNTR lengths of 32,138 loci in genome $g$; $\mathrm{kms}_s$ is the sum of $k$-mer counts in each locus mapped by sample $s$; $\mathrm{cov}_s$ is the global read depth of sample $s$ estimated by averaging the read depths of 397 unique regions without any types of repeats or duplications.

The ground truth VNTR length of a locus $l$ in genome $g$ is averaged across haplotypes:

$$L_{g,l} = \frac{1}{H} \sum_{h=1}^{H} L_{g,h,l} \qquad (2)$$

where $H$ is the number of haplotype(s) in genome $g$, i.e., 2 for normal individuals and 1 for complete hydatidiform mole (CHM) samples.

With the above bias terms, the VNTR length of locus $l$ in sample $s$ can be computed by:

$$L_{s,l} = \frac{\mathrm{kms}_{s,l}}{\mathrm{cov}_s \times b_{\hat{s}}} \qquad (3)$$

where $\mathrm{cov}_s$ is same as described above; $b_{\hat{s}}$ is the estimated LSBs computed from sample $\hat{s}$ with ground truth VNTR lengths; $\mathrm{kms}_{s,l}$ is the sum of $k$-mer counts of locus $l$ mapped by sample $s$. We assume the LSBs that best approximates $b_s$ come from samples within the same sequencing run. Without prior knowledge on the ground truth VNTR lengths of $s$ and therefore $b_s$, we determine the "closest" sample $\hat{s}$ w.r.t. $s$ based on $r^2$ between the read depths, **RD**, of the 397 unique regions as follows:

$$\hat{s} = \underset{s',s' \in \mathrm{GT}, s' \neq s}{\mathrm{argmax}} \; r^2(\mathbf{RD}_{s'}, \mathbf{RD}_s) \qquad (4)$$

where GT is the set of samples with ground truths and within the same sequencing run as $s$. We cross-validate our approach by leaving one sample out of the pan-genome database and evaluating the prediction accuracy on the excluded sample.

For comparison, VNTR lengths were also estimated by a read depth method. For each VNTR region, the read depth, computed with samtools bedcov -j, was divided by the global read depth, computed from the 397 nonrepetitive regions, to give the length estimate.

*Comparing with GraphAligner.* The compact de Bruijn graph of each VNTR locus was generated with bcalm v2.2.3 (option: -kmer-size 21 -abundance-min 1) using the VNTR sequences from all assemblies as input.

GFA files were then reindexed and concatenated to generate the RPGGs for 32,138 loci. Error-free paired-end reads were simulated from all VNTR regions at 2x coverage with 150 bp read length and 600 bp insert size (300 bp gap between each end). Reads were aligned to the RPGG using GraphAligner v1.0.11 with option -x dbg --seeds-minimizer-length 21. Reads with alignment identity >90% were counted from the output gam file. To compare in a similar setting, danbing-tk was run with option -gc -thcth 117 -k 21 -cth 45 -rth 0.5 to assert >90% identity for all reads aligned, given that (read_length − kmer_size + 1) × 0.9 = 117.

## Population analysis

*$V_{\mathrm{ST}}$ calculation.* $V_{\mathrm{ST}}$ was calculated according to Redon et al.[41]:

$$V_{\mathrm{ST}}[i] = \max\left(0, \frac{\mathrm{var}_{\mathrm{all}} - \frac{1}{n}\sum_{p \in P}\mathrm{var}_p \times n_p}{\mathrm{var}_{\mathrm{all}}}\right) \qquad (5)$$

Top $V_{\mathrm{ST}}$ loci were considered as the sites with $V_{\mathrm{ST}}$ at least three standard deviations above the mean.

*Properties of $V_{\mathrm{ST}}$.* The dosage of a VNTR for an individual $j$ is $x_j \geq 0$. Consider $n_{\mathrm{T}}$ individuals consisting of $P$ populations, with $n_i$ individuals each, with population mean and variance, $\mu_i$ and $\sigma_i^2$, and a global mean and variance $\mu_{\mathrm{T}}$ and $\sigma_{\mathrm{T}}^2$ for all individuals. Population stratification is calculated as $V_{\mathrm{ST}} = \frac{1}{\sigma_{\mathrm{T}}^2}\left(\sigma_{\mathrm{T}}^2 - \frac{\sum \sigma_i^2 n_i}{\sum n_i}\right)$.

The mean across populations, $\mu_{\mathrm{T}}$ is calculated as $(\sum \mu_i n_i)/\sum n_i$. The variance is $E(x_j - \mu_{\mathrm{T}})^2$ for all individuals, and this may be separated out by population as $\sum_i E(x_k^i - \mu_{\mathrm{T}})^2 n_i / \sum_i n_i$, using $x_k^i$ to denote the $k$th individual in population $i$. The value $\sum_i E(x_k^i - \mu_{\mathrm{T}})^2$ may be computed as:

$$E(x_k^i - \mu_{\mathrm{T}})^2 = E\left((x_k^i - \mu_i) + (\mu_i - \mu_{\mathrm{T}})\right)^2$$

$$= E\left((x_k^i - \mu_i)^2 + 2(x_k^i - \mu_i)(\mu_i - \mu_{\mathrm{T}}) + (\mu_i - \mu_{\mathrm{T}})^2\right)$$

since $E(x_k^i) = \mu_i, E(x_k^i - \mu_i) = E(x_k^i) - E(\mu_i) = 0$

$$= \sigma_i^2 + E(\mu_i - \mu_{\mathrm{T}})^2$$

$$= \sigma_i^2 + (\mu_i - \mu_{\mathrm{T}})^2 \qquad (6)$$

The total population variance $\sigma_{\mathrm{T}}^2$ relative to the population mean, variance, size, and global mean is:

$$\sigma_{\mathrm{T}}^2 = \frac{\sum_i n_i\left(\sigma_i^2 + (\mu_i - \mu_{\mathrm{T}})^2\right)}{\sum_i n_i} \qquad (7)$$

Replacing this in the calculation of $V_{\mathrm{ST}}$ gives:

$$V_{\mathrm{ST}} = \frac{1}{\sigma_{\mathrm{T}}^2}\left(\frac{\sum_i n_i\left(\sigma_i^2 + (\mu_i - \mu_{\mathrm{T}})^2\right)}{\sum_i n_i} - \frac{\sum_i \sigma_i^2 n_i}{\sum_i n_i}\right) = \frac{\sum_i n_i(\mu_i - \mu_{\mathrm{T}})^2}{n_{\mathrm{T}} \times \sigma_{\mathrm{T}}^2} \qquad (8)$$

*Identifying unstable loci.* A locus was annotated as a candidate for being unstable if at least one individual had outlying $k$-mer dosage $\geq$ six standard deviations above the mean, using population and locus-specific summary statistics on data discarding individuals with dosage <10 or a bimodal distribution was not detected (diptest v0.75–7, $P > 0.9$). Among this set, the number of times each genome

appeared as an outlier was used to select a set of genomes with an over abundant contribution to fragile loci. Any candidate locus with an individual that was an outlier in at least four other loci was removed from the candidate list. The loci were compared to gencode v34, excluding readthrough, pseudogenes, noncoding RNA, and nonsense transcripts.

*Identifying differential motif usage and expansion.* Sample outliers in the 1000 Genomes were detected from the LSBs over 397 control regions and the VNTR dosages over 32,138 loci using DBSCAN. A total of 119/2504 samples were removed from downstream analysis. We use the EAS population as the reference for measuring differential motif usage and expansion. Initially, a lasso fit using the statsmodels.api.OLS function in python statsmodel v0.10.1[55] was performed for each locus to identify the $k$-mer with the most variance explained (VEX) in VNTR lengths using the following formula: $\mathbf{y} = X\mathbf{b} + \boldsymbol{\epsilon}$, where $\mathbf{y} \in \mathbb{R}^N$ is the VNTR length of $N$ individuals in the EAS population; $X \in \mathbb{R}^{N \times M}$ is the $k$-mer dosage matrix for $N$ individuals with $M$ $k$-mers; $\mathbf{b} \in \mathbb{R}^N$ is the model coefficient, and $\boldsymbol{\epsilon} \sim N(0, \sigma^2)$ is the error term. The lasso penalty weight $\alpha$ was scanned starting at 0.9 with at a step size of $-0.1$ until at least one covariate has a positive weight or $\alpha$ is below 0.1. The $k$-mer with the highest weight is denoted as the most informative $k$-mer (mi-kmer) for the locus.

To identify loci with differential motif usage between populations, we subtracted the median count of the mi-kmer of the AFR from the EAS population for each locus, denoted as $\mathrm{kmc}_d$. The null distribution of $\mathrm{kmc}_d$ was estimated by bootstrap. Specifically, EAS individuals were sampled with replacement $N_{\mathrm{EAS}} + N_{\mathrm{AFR}}$ times, matching the sample sizes of the EAS and AFR populations, respectively. The bootstrap statistics, $\mathrm{kmc}_d^*$, were computed by subtracting the median count of the mi-kmer of the last $N_{\mathrm{AFR}}$ from the first $N_{\mathrm{EAS}}$ bootstrap samples for each locus. The estimated null distribution is then used to determine the threshold for calling a locus having significant differential motif usage between populations (two-sided $P < 0.01$).

To identify loci with differential motif expansion between populations, we subtracted the proportion of VEX by mi-kmer in the AFR from the EAS population, denoted as $r_d^2$. The null distribution of $r_d^2$ was estimated by bootstrap in a similar sampling procedure as $\mathrm{kmc}_d$, except for subtracting the proportion of VEX by the mi-kmer in the last $N_{\mathrm{AFR}}$ from the first $N_{\mathrm{EAS}}$ bootstrap samples for each locus. The estimated null distribution is used to determine the threshold for calling a locus having significant differential motif expansion between populations (two-sided $P < 0.01$).

## eQTL mapping

*Retrieving datasets.* WGS datasets of 879 individuals, normalized gene expression matrices and covariates of all tissues are accessed from the GTEx Analysis V8 (dbGaP Accession phs000424.v8.p2).

*Genotype data preprocessing.* VNTR lengths are genotyped using daunting-tk with options: -gc -thcth 50 -cth 45 -rth 0.5. All the $k$-mer counts of a locus are summed and adjusted by global read depth and ploidy to represent the approximate length of a locus. Sample outliers were detected from the LSBs over 397 control regions and the VNTR dosages over 32,138 loci using DBSCAN. A total of 26/838 samples were removed from downstream analysis. Adjusted values are then $z$-score normalized as input for eQTL mapping.

*Expression data preprocessing.* The downloaded expression matrices are already preprocessed such that outliers are rejected and expression counts are quantile normalized as standard normal distribution. Confounding factors such as sex, sequencing platform, amplification method, technical variations and population structure are removed prior to eQTL mapping to avoid spurious associations. Technical variations are corrected with the covariates, including PEER factors, provided by the GTEx Consortium. Population structures are corrected with the top 10 principal components (PCs) from the SNP matrix of all samples. Particularly, principal component analysis (PCA) was performed jointly on the intersection of the SNP sets from GTEx samples and 1KGP Omni 2.5 SNP genotyping arrays (ftp://ftp.1000genomes.ebi.ac.uk/vol1/ftp/release/20130502/supporting/hd_genotype_chip/ALL.chip.omni_broad_sanger_combined.20140818.snps.genotypes.vcf.gz). This is done by first using CrossMap v0.4.0 to liftover the SNP sites from Omni 2.5 arrays to GRCh38, followed by extracting the intersection of the two SNP sets using vcftools isec. The SNP set is further reduced by LD-pruning with plink v1.90b6.12 using the options: --indep 50 5 2, leaving a total of 757,000 sites. Finally, PCA on the joint SNP matrix was done by smartpca v13050. The normalized expression matrix are residualized with the above covariates using the following formula:

$$Y = (I - H)Y' \qquad (9)$$

$$H = C(C^T C)^{-1} C^T \qquad (10)$$

where $Y$ is the residualized expression matrix; $Y'$ is the normalized expression matrix; $H$ is the projection matrix; $I$ is the identity matrix; $C$ is the covariate matrix

where each column corresponds to a covariate mentioned above. The residualized expression values are $z$-score normalized as the input of eQTL mapping.

*Association test.* VNTRs within 100 kb to a gene are included for eQTL mapping. Linear regression was done using the statsmodel.api.OLS function in python statsmodel v0.10.1[55] with expression values as the dependent variable and genotype values as the independent variable. Nominal *P*-values are computed by performing *t*-tests on slope. Adjusted *P*-values are computed by Bonferroni correction on nominal *P*-values. Under the assumption of at most one causal VNTR per gene, we control gene-level false-discovery rate at 5%. Specifically, the adjusted *P*-values of the lead VNTR for each gene are taken as input for Benjamini–Hochberg procedure using statsmodels.stats.multitest.fdrcorrection v0.10.1. Lead VNTRs that passed the procedure are identified as eVNTRs.

**Reporting summary**. Further information on research design is available in the Nature Research Reporting Summary linked to this article.

## Data availability

Data accession IDs are given in Supplementary Table 4. Six HGSVC diploid assemblies that support the finding of this study are available at https://www.internationalgenome.org/data-portal/data-collection/hgsvc2. The remaining 13 diploid assemblies, the RPGG data structure and the *k*-mer dosage table for 1KGP samples are available at https://doi.org/10.5281/zenodo.4758205. The whole-genome sequencing data of 1KGP samples (PRJEB36890) are available at https://www.internationalgenome.org/data-portal/data-collection/30x-grch38. The following cell lines/DNA samples were obtained from the NIGMS Human Genetic Cell Repository at the Coriell Institute for Medical Research: [NA06984, NA06985, NA06986, NA06989, NA06991, NA06993, NA06994, NA06995, NA06997, NA07000, NA07014, NA07019, NA07022, NA07029, NA07031, NA07034, NA07037, NA07045, NA07048, NA07051, NA07055, NA07056, NA07340, NA07345, NA07346, NA07347, NA07348, NA07349, NA07357, NA07435, NA10830, NA10831, NA10835, NA10836, NA10837, NA10838, NA10839, NA10840, NA10842, NA10843, NA10845, NA10846, NA10847, NA10850, NA10851, NA10852, NA10853, NA10854, NA10855, NA10856, NA10857, NA10859, NA10860, NA10861, NA10863, NA10864, NA10865, NA11829, NA11830, NA11831, NA11832, NA11839, NA11840, NA11843, NA11881, NA11882, NA11891, NA11892, NA11893, NA11894, NA11917, NA11918, NA11919, NA11920, NA11930, NA11931, NA11932, NA11933, NA11992, NA11993, NA11994, NA11995, NA12003, NA12004, NA12005, NA12006, NA12043, NA12044, NA12045, NA12046, NA12056, NA12057, NA12058, NA12144, NA12145, NA12146, NA12154, NA12155, NA12156, NA12234, NA12239, NA12248, NA12249, NA12264, NA12272, NA12273, NA12274, NA12275, NA12282, NA12283, NA12286, NA12287, NA12329, NA12335, NA12336, NA12340, NA12341, NA12342, NA12343, NA12344, NA12347, NA12348, NA12375, NA12376, NA12383, NA12386, NA12399, NA12400, NA12413, NA12414, NA12485, NA12489, NA12546, NA12707, NA12708, NA12716, NA12717, NA12718, NA12739, NA12740, NA12748, NA12749, NA12750, NA12751, NA12752, NA12753, NA12760, NA12761, NA12762, NA12763, NA12766, NA12767, NA12775, NA12776, NA12777, NA12778, NA12801, NA12802, NA12812, NA12813, NA12814, NA12815, NA12817, NA12818, NA12827, NA12828, NA12829, NA12830, NA12832, NA12842, NA12843, NA12864, NA12865, NA12872, NA12873, NA12874, NA12875, NA12877, NA12878, NA12889, NA12890, NA12891, NA12892]. The whole-genome sequencing and expression data of GTEx samples (phs000424.v8.p2) can be accessed from https://www.gtexportal.org/home/datasets. The Source Data for Figs. 1c, 2b, c, e, 3, 4e, Supplementary Figs. 1, 5–17, 19-20, 29, 32, 34–40 and Supplementary Table 6 are available at https://doi.org/10.5281/zenodo.4758205. Source data are provided with this paper.

## Code availability

The overall analysis pipeline is delivered in a software package at https://github.com/ChaissonLab/danbing-tk[56].

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

## Acknowledgements

This work is funded by NHGRI U01 HG010973 and U24 HG007497. 1000 Genomes Acknowledgement: the cell lines/DNA samples obtained from the NIGMS Human Genetic Cell Repository at the Coriell Institute for Medical Research were generated at the New York Genome Center with funds provided by NHGRI Grants 3UM1HG008901-03S1 and 3UM1HG008901-04S2.

## Author contributions

T.Y.L. and M.J.P.C. performed data analysis and wrote the manuscript. M.J.P.C. supervised the work. The Human Genome Structural Variation Consortium generated sequencing data.

## Competing interests

The authors declare no competing interests.

## Additional information

## The Human Genome Structural Variation Consortium

Katherine M. Munson[2], Alexandra P. Lewis[2], Qihui Zhu[3], Luke J. Tallon[4], Scott E. Devine[4], Charles Lee[3,5,6] & Evan E. Eichler[2,7]

[2]Department of Genome Sciences, University of Washington School of Medicine, Seattle, WA, USA. [3]The Jackson Laboratory for Genomic Medicine, Farmington, CT, USA. [4]Institute for Genome Sciences, University of Maryland School of Medicine, Baltimore, MD, USA. [5]Precision Medicine Center, The First Affiliated Hospital of Xi'an Jiaotong University, Xi'an, Shaanxi, China. [6]Department of Graduate Studies–Life Sciences, Ewha Womans University, Ewhayeodae-gil, Seodaemun-gu, Seoul, South Korea. [7]Howard Hughes Medical Institute, University of Washington, Seattle, WA, USA.

