## [Peer Review File · Nature Communications]

REVIEWER COMMENTS

Reviewer #1 (Remarks to the Author):

The authors present a fascinating study of VNTR variation, which they access using a relatively novel analysis approach. Following recent work in pangenomic methods, they utilize repeat de Bruijn graphs as pangenome models (RPGGs) to describe variation in VNTR loci in samples with short read sequencing. To align read data, they develop a custom heuristic alignment method based on the exact matching of reads to walks through these graphs. Each sample is represented as a normalized coverage across the feature space of these graphs, where features are kmers or nodes in the RPGG. By embedding VNTR variation found in high-quality assembled genomes within their reference data structure, the authors are able to elucidate the nature of genomic variation in these dynamic regions and its impact on genome function.

This work struck me as a very sound and straightforward take on an aspect of genome variation that is frequently overlooked due to the complexity of analysis of it. They also have the opportunity to build on human assemblies based on long reads to build their reference structure, which is a critical requirement for such work.

I have a few concerns with the approach the authors took, and some confusion about important specifics of their method.

I was not clear if the RPGG included the entire reference. If so, how did the authors handle mapping to non-VNTR regions? At the chosen kmer sizes, the graph of a human genome would be extremely tangled. If they do not map to the entire human reference, then mismapping of reads from the rest of the genome into the RPGGs could introduce strange artifacts. I apologize if this was made clear in the text, but I did not find it immediately obvious.

The authors note, "There was little effect on downstream analysis for values of k between 17 and 25, and so $k=21$ was used for all applications." This left me curious: what happens at higher k ? My experience with building DBGs from such regions suggests that the kmers that are used are extremely low, and likely to generate very tangled structures that might incorporate pieces of surrounding sequence. As the authors' overall result is likely to be robust to this (they are projecting their samples into this feature space and comparing them uniformly within it) this may be unimportant. But I was left very curious what happens at $k=31$, $k=47$, $k=63$ and so forth. The statement suggests that the results change, but how?

I found an error in the authors' assessment of existing methods to align reads to graphs. They state "While several methods exist to find alignments that do not reuse cycles (Garrison et al. 2018; Rakocevic et al. 2019)..." However, the cited article (Garrison 2018) specifically demonstrates alignment to graphs with cycles and arbitrary structural variation (For clarity, I am the author.) One of the key experiments demonstrates that long PacBio reads map better to a whole genome alignment graph of seven *S. cerevisiae* strains that contains very complex structural variation than they do to a linear assembly from the

standard reference strain. That is not to say that the method is optimal, as it is around an order of magnitude slower than GraphAligner, as shown in (Rautiainen, Mäkinen, and Marschall 2019). I suggest the authors rephrase this to indicate the shortfall while remaining precise about their citation, something like “While methods exist to find alignments that do not reuse cycles (Rakocevic et al. 2019), others allow alignment to cyclic graphs, but with high computational costs when the graphs are of the type we build (Garrison 2018).”

I did not find the comparison with GraphAligner or other methods to be very convincing. Perhaps it should simply be removed. It does not add anything and gives the impression that the authors weren't very careful. One key issue is that the authors do not verify if GraphAligner or their method align the simulated reads to the correct location in the graph they were simulated from. They simply note that GraphAligner yielded alignments at lower than expected identity. Could these have been mapped correctly, while those from danbing-tk were mapped erroneously, with a high reported identity? The authors cannot exclude this. Also, are the settings applied to GraphAligner those recommended by the author for the kinds of graphs that they build? This should be indicated.

In reflection, I worry that the authors may overstate the technical advantage of their approach. Was it even necessary to use a DBG, or could they have just projected their read sequences into a set of kmer counts? Do the authors feel they have explained the advantage of their model over something simpler like this? This aspect can simply be minimized. The authors have been pragmatic, and built a focused solution for their application, which is the exploration of VNTRs in the human population. It is not necessary for them to exhaustively compare with other tools, and an incomplete comparison is perhaps more confusing to readers than nothing at all, and the space could be better spent on a more thorough evaluation of the accuracy of their own method (e.g. a ROC curve of the alignment accuracy versus mapping quality).

Overall, the authors have made a significant contribution to what will certainly be a rapidly developing area of study in coming years. I hope they find my comments helpful and continue their work on this topic.

Erik Garrison

Reviewer #3 (Remarks to the Author):

The authors have responded to the comments from the review in a comprehensive manner. In response to one of the comments, they have updated the method for defining VNTR boundaries which has improved the accuracy of the genotyping and has also resulted in a substantial change in the results. Specifically, many of the highlighted loci and eVNTRs have changed. This suggests that the robustness of the method can be improved. For example, if the authors were to use a different set of LRS assemblies to build the pangenome graphs, it is feasible that the set of VNTRs that can be genotyped with high confidence will change a lot.

Also, in response to comment #29 (from the response document), the authors state that "As the number of included genomes increases, the probability of the boundary in one genome not aligning with others also increases." This indicates that the process for selecting VNTR

boundaries is not robust to outliers and can adversely affect the genotyping accuracy. Does this imply that VNTR pangenome graphs built using hundreds of LRS assemblies (that may be available in the future) will actually perform worse? This problem seems similar to what has been observed in the context of read alignment to graph genomes where adding more variants to the graph can actually be detrimental to the alignment accuracy (see <https://genomebiology.biomedcentral.com/articles/10.1186/s13059-018-1595-x>). Overall, this is a comprehensive piece of work on addressing an under-studied source of variation in human genomes. However, the robustness of the method (i.e. the VNTR genotyping) is still of some concern to me. Hopefully, this can be addressed in future work.

Minor comment:

I also looked at the new VNTR eQTLs reported in the paper. For the ERAP2 locus, the authors state that "This VNTR is a unique sequence in GRCh38 that is a 101 bp tandem duplication in 17/38 of the haplotypes." Does this mean that the VNTR is simply a bi-allelic variant or are there more than two alleles of the VNTR? To demonstrate that the VNTR is driving the change in gene expression, it may be feasible to test if there is SNV at the same locus that is also associated with the gene expression and if the association is weaker than the VNTR.

Reviewer #4 (Remarks to the Author):

The authors have performed an extensive analysis to address comments from reviewers. VNTR, as one of the most divergent and under-studied categories of genomic variants, is very difficult to be called and to be benchmarked. The authors' additional analysis is solid. With the following minor issues addressed, the manuscript should be good to be published.

1. The term "read sampling bias" is not clearly explained with LSB. Although the authors have changed "read sampling bias" in Figure 3a, I recommend the authors review this term across the full text and supplementary figures. Specify/replace it if needed.

2. To address the reviewer comment 32, the authors added a figure showing that sampling bias is a reason for the genotyping variation in Figure 3(a). If this indicates data quality as the reason for this variation, the authors could state that.

Additionally, there is a better 30x genome of NA24385 at ENA PRJEB35491. If the authors would like to solve the quality problem of NA24385 they can redo their benchmarking with this data.

3. Figure 3(a) right plot indicates variation per-locus in the three methods as well. It could be helpful for the authors to add some text and/or cases to discuss a few poorly genotyped locus and show why they do not work well in one method or all methods. The authors have already shown some factors in Supplementary Figure 28 & 29 (TR length and GC). Overall, a brief discussion or case analysis for VNTRs that do not work well with RPKG will help people to understand how difficult the VNTR calling is and to understand your method better.

4. Reviewer comment 18 asks for the purity cutoff in TRF to be discussed. e.g. if a repeat is

>99% pure and is short enough, it can be genotyped by GangSTR, too. Better to quantify how many of your VNTRs are like this.

Text comments:

1. On page 2 line 4, "single-molecule sequencing" should be "long-read sequencing" to match the abbreviation "LRS".
2. On page 4 line 12, "sequenced by either PacBio single long read (SLR)": I recommend replacing "SLR" with "contiguous long read" (CLR). This is the term that PacBio is officially using.
3. On page 6 line 6, the authors mentioned that alignment with cycles is "recently solved by GraphAligner". Actually, the sequence-graph version of ExpansionHunter (2019) also aligns reads to a local cycled graph. It could be nice to include this as well.

Reviewer #1 (Remarks to the Author):

The authors present a fascinating study of VNTR variation, which they access using a relatively novel analysis approach. Following recent work in pangenomic methods, they utilize repeat de Bruijn graphs as pangenome models (RPGGs) to describe variation in VNTR loci in samples with short read sequencing. To align read data, they develop a custom heuristic alignment method based on the exact matching of reads to walks through these graphs. Each sample is represented as a normalized coverage across the feature space of these graphs, where features are kmers or nodes in the RPGG. By embedding VNTR variation found in high-quality assembled genomes within their reference data structure, the authors are able to elucidate the nature of genomic variation in these dynamic regions and its impact on genome function.

This work struck me as a very sound and straightforward take on an aspect of genome variation that is frequently overlooked due to the complexity of analysis of it. They also have the opportunity to build on human assemblies based on long reads to build their reference structure, which is a critical requirement for such work.

I have a few concerns with the approach the authors took, and some confusion about important specifics of their method.

Comment 1:

I was not clear if the RPGG included the entire reference. If so, how did the authors handle mapping to non-VNTR regions? At the chosen kmer sizes, the graph of a human genome would be extremely tangled.

Response 1:

The manuscript states that the RPGG is constructed on separate VNTR sequences, indicating that the whole reference is not used:

The RPGGs are constructed as disjoint bi-directional de Bruijn graphs of each VNTR locus and flanking 700 bases from the haplotype-resolved assemblies.

To make the description more clear, in the context of evaluating mapping accuracy including reads from elsewhere in the genome, we have modified the statement:

Using danbing-tk, 99.997% of VNTR-simulated reads were aligned with >90% identity. When reads from the entire genome are considered, for 96.6% of the loci (71,080/73,582), danbing-tk can map >90% of the reads back to their original VNTR regions.

to:

Using danbing-tk, 99.997% of VNTR-simulated reads were aligned with >90% identity. **The RPGG is only built on VNTRs and their flanking sequences, excluding the rest of the genome.** When reads from the entire genome are considered, for 96.6% of the loci (71,080/73,582), danbing-tk can map >90% of the reads back to their original VNTR regions.

Comment 1 (continued): If they do not map to the entire human reference, then mismapping of reads from the rest of the genome into the RPGGs could introduce strange artifacts. I apologize if this was made clear in the text, but I did not find it immediately obvious.

Response 1 (continued):

In our submission, we had quantified the extent of mismapping reads into the RPGG from the rest of the genome, as stated:

Misaligned reads from either other VNTR loci or untracked regions target relatively few loci; 3.6% (2,635/73,582) loci have at least one read misaligned from outside the locus.

We have updated the text to more explicitly characterize how this quantifies mapping artifacts:

Misaligned reads from either other VNTR loci **included in the RPGG or the remainder of the genome not included in the RPGG** target relatively few loci; 3.6% (2,635/73,582) loci have at least one read misaligned from outside the locus.

Comment 2:

The authors note, “There was little effect on downstream analysis for values of k between 17 and 25, and so $k=21$ was used for all applications.” This left me curious: what happens at higher k ? My experience with building DBGs from such regions suggests that the kmers that are used are extremely low, and likely to generate very tangled structures that might incorporate pieces of surrounding sequence. As the authors’ overall result is likely to be robust to this (they are projecting their samples into this feature space and comparing them uniformly within it) this may be unimportant. But I was left very curious what happens at $k=31$, $k=47$, $k=63$ and so forth. The statement suggests that the results change, but how?

Response 2:

Because the de Bruijn graphs are constructed only on VNTR loci and their surrounding sequences (see response 1), k -mers from distal regions of the genome will not be “tangled” between separate locus-RPGGs. Furthermore, as described in our methods, we use our boundary-expansion algorithm to account for the local sequence context of a VNTR where the boundaries of the VNTR and surrounding sequence are not well defined (e.g. accounting for pieces of surrounding sequence).

Furthermore, as we previously responded to this comment, danbing-tk uses de Bruijn graphs to represent the repeat structure rather than the linear sequence of a VNTR (or shared linear orthologous sequences, as commonly done with pangenome graphs). We refer to the additional details from our previous response below:

The aim of danbing-tk is to accurately represent the repeat structure of VNTR sequences, while enabling accurate mapping of sequences into each VNTR locus. While larger values of k

can help produce less “tangled” graphs, motifs that are smaller than the length of the k-mer will be represented multiple times in the de Bruijn graph. Furthermore, as k increases the pangenome graph will become similar to a variation graph without merging of similar sequences. This is exactly the problem for representing pangenomes that danbing-tk solves using repeat graphs (e.g. de Bruijn with smaller k). Our aim is to enable measuring sample-specific motif usage, which will be hampered the larger the value of k.

We have modified the manuscript to clarify how the RPGG is constructed to highlight how incorporating pieces of surrounding sequence is not a factor. Specifically, we have modified the statement:

The RPGGs are constructed as disjoint bi-directional de Bruijn graphs of each VNTR locus and flanking 700 bases from the haplotype-resolved assemblies.

to:

The RPGGs **are a collection of independently constructed** bi-directional de Bruijn graphs of each VNTR locus and flanking 700 bases from the haplotype-resolved assemblies.

We also have added the statement:

The RPGG differs from a standard bi-directional de Bruijn graph because a *k*-mer may be repeated in multiple subgraphs.

Comment 3:

I found an error in the authors’ assessment of existing methods to align reads to graphs. They state “While several methods exist to find alignments that do not reuse cycles (Garrison et al. 2018; Rakocevic et al. 2019)...” However, the cited article (Garrison 2018) specifically demonstrates alignment to graphs with cycles and arbitrary structural variation (For clarity, I am the author.) One of the key experiments demonstrates that long PacBio reads map better to a whole genome alignment graph of seven *S. cerevisiae* strains that contains very complex structural variation than they do to a linear assembly from the standard reference strain. That is not to say that the method is optimal, as it is around an order of magnitude slower than GraphAligner, as shown in (Rautiainen, Mäkinen, and Marschall 2019). I suggest the authors rephrase this to indicate the shortfall while remaining precise about their citation, something like “While methods exist to find alignments that do not reuse cycles (Rakocevic et al. 2019), others allow alignment to cyclic graphs, but with high computational costs when the graphs are of the type we build (Garrison 2018).”

Response 3:

We have updated our manuscript with text similar to what was recommended, and also reference ExpansionHunter:

While methods exist to find alignments that do not reuse cycles (Rakocevic et al. 2019), others allow alignment to cyclic graphs but with high computational costs when applied to RPGG (Garrison et al. 2018) or are limited to alignment in STR regions (Dolzhenko et al. 2019).

We defend our initial statement as a difference in semantics for cycle-reuse in graph alignments, based on the text from the methods section from the vg manuscript (reproduced below). This text along with the supplemental figures demonstrate a transformation of a cyclical topology to a directed-acyclic graph in which paths from the original graph are limited by expansion parameter in the software:

To avoid the complications introduced by cycles and inversions, we transform the local graph region into a directed acyclic graph (DAG) while maintaining an embedding in the original, potentially cyclic bidirected graph (Supplementary Figs. 3 and 4) (Garrison, E., *et al.*, 2018, Nature Biotechnology)

Comment 4:

I did not find the comparison with GraphAligner or other methods to be very convincing. Perhaps it should simply be removed. It does not add anything and gives the impression that the authors weren't very careful. One key issue is that the authors do not verify if GraphAligner or their method align the simulated reads to the correct location in the graph they were simulated from. They simply note that GraphAligner yielded alignments at lower than expected identity. Could these have been mapped correctly, while those from danbing-tk were mapped erroneously, with a high reported identity? The authors cannot exclude this.

Response 4:

We reanalyzed the alignment results from danbing-tk and GraphAligner in a whole-genome error-free simulation and classified the alignments as follows (Supplementary Table 1). Since GraphAligner does not consider paired-end information, read pairs can fail to map with high identity (10.5%), split-align to different loci (3.2%) or have a missing mate (3.3%). This drops the percentage of read pairs mapped to 95.9% in GraphAligner versus 99.96% in danbing-tk, and the percentage of correctly mapped read pairs to 81.9% in GraphAligner versus 99.62% in danbing-tk.

Supplementary Table 1. Comparison of alignment statistics between danbing-tk and GraphAligner.

	danbing-tk	GraphAligner
Read pairs mapped	258516 (99.96%)	247930 (95.9%)
Read pairs correctly mapped	257638 (99.62%)	211919 (81.9%)
Read pairs mismapped	878 (0.34%)	532 (0.21%)
Read pairs with low identity in	0 (0%)	27259 (10.5%)

at least one end		
Read pairs split	0 (0%)	8220 (3.2%)
Singletons	0 (0%)	8629 (3.3%)
Loci with correct read pairs	28468 (98.5%)	28405 (98.3%)

We believe that this analysis is convincing to show the performance gap between danbing-tk and GraphAligner.

Comment 5:

Also, are the settings applied to GraphAligner those recommended by the author for the kinds of graphs that they build? This should be indicated.

Response 5:

We have communicated with the authors of GraphAligner. They suggested trying various seeding parameters including `--seeds-mem-count`, `--seeds-mxm-length`, `--try-all-seeds` and `--bandwidth`. Based on our experience, these parameters can moderately improve the alignment quality but the most important factor is `--seeds-minimizer-length`, which was set to 21 due to the k -mer size of the input de Bruijn graph.

Comment 6:

In reflection, I worry that the authors may overstate the technical advantage of their approach. Was it even necessary to use a DBG, or could they have just projected their read sequences into a set of k mer counts? Do the authors feel they have explained the advantage of their model over something simpler like this? This aspect can simply be minimized.

Response 6:

It is highly necessary to use a DBG. Projecting read sequences into a k -mer set is implemented in the first step of our mapping algorithm. To be suitably sensitive, it is necessary to not require all k -mers from a read to align (e.g. accounting for genetic divergence and sequencing error). One could rank loci by the number of k -mers shared with a read, and select the top-ranked locus for the alignment. However, even when selecting for the top-ranked locus using k -mers, too many reads incorrectly map and produce abundant false positives. On the other hand, it is also not practical to set a high threshold for the first step just to avoid false positives since sequencing errors and sequence divergence are common in VNTR regions. We therefore added the threading step to reduce both false positive rate (FPR) and false negative rate (FNR=1-TPR) by leveraging the graph information to correct errors or detect divergence (Supplementary Fig. 38, left panel). On 1000 Genomes and GTEx

datasets, ~40% of reads entered the first step are removed in the threading step (e.g. they likely came from other regions of the genome and spuriously map to VNTR sequences based on shared k -mer counts).

Below, we demonstrate the TPR/FPR of alignment when considering shared k -mers only (w/o threading), and with threading. This has been added to the supplementary material.

Supplementary Figure 38 (left panel). Comparing the alignment accuracy with and without threading. Paired-end 150 bp reads were simulated with or without SNVs and mapped to unpruned RPGG. A read is considered correctly mapped if its VNTR k -mers are assigned to the correct VNTR locus. Each curve is parameterized by percent identity threshold (linspace distributed between 35% and 90%). For runs with threading enabled, `cth` was set to 30, and four nucleotide corrections were allowed. TPR, true positive rate; FPR, false positive rate.

Comment 7:

The authors have been pragmatic, and built a focused solution for their application, which is the exploration of VNTRs in the human population. It is not necessary for them to exhaustively compare

with other tools, and an incomplete comparison is perhaps more confusing to readers than nothing at all, and the space could be better spent on a more thorough evaluation of the accuracy of their own method (e.g. a ROC curve of the alignment accuracy versus mapping quality).

Response 7:

The figure below emphasizes our previous evaluation with threading enabled (Supplementary Fig. 38, right panel). When there is no sequencing error or divergence, >99% of VNTR reads can be mapped at 90% identity, while >96% can be mapped at 85% identity when there is a single SNV in each read, i.e. two for a read pair.

Supplementary Figure 38 (right panel). Evaluation of simulated read alignments. Paired-end 150 bp reads were simulated with or without SNVs and mapped to unpruned RPGG with threading enabled. A read is considered correctly mapped if its VNTR k -mers are assigned to the correct VNTR locus. Each curve is parameterized by percent identity threshold (linspace distributed between 35% and 90%). The `-cth` option was set to 30 while allowing for four nucleotide corrections. TPR, true positive rate; FPR, false positive rate.

Comment 8:

Overall, the authors have made a significant contribution to what will certainly be a rapidly developing area of study in coming years. I hope they find my comments helpful and continue their work on this topic.

Response 8:

We thank the reviewer for his comments. The suggestions did help us improve our manuscript and deliver a clearer message to the readers.

Erik Garrison

Reviewer #3 (Remarks to the Author):

The authors have responded to the comments from the review in a comprehensive manner. In response to one of the comments, they have updated the method for defining VNTR boundaries which has improved the accuracy of the genotyping and has also resulted in a substantial change in the results.

Comment 9:

Specifically, many of the highlighted loci and eVNTRs have changed. This suggest that the robustness of the method can be improved.

Response 9:

We acknowledge that the change in highlighted loci would cause concern for the robustness of the method, however while the accuracy of the method was improved for the first revision, the majority of changes were due to reporting on a different set of loci between revisions. In each revision, the loci that are included in the RPGG are based on an alignment quality assessment using matched short and long-read genomes. Our method development in the first revision resulted in a non-uniform improvement in alignment quality, enabling new loci to be included in the RPGG compared to the initial submission. To limit computational burden, we chose to analyze a similar number of loci as the initial submission, which resulted in a different set of loci. To demonstrate robustness of the method, we used improvements in alignment efficiency to analyze the complete set of 73,582 VNTR loci that may be defined from the long read sequence assemblies. This allows us to evaluate the changes for 90.1% (26,182/29,052) of the loci from the first submission. We include the Vst and eQTL analysis of the expanded dataset in our supplementary material, but keep the loci reported on in the main manuscript the same as in the initial revision in order to focus on those with very high alignment quality. This analysis is presented below, and includes:

1. Reproduction of population stratification using a separate set of genomes sequenced by the 1000-Genomes project.
2. Shown correlation of population stratification between revisions of our method.
3. Demonstrated stability of pangenome-graph construction with respect to inclusion of additional genomes.
4. Shown a tight agreement between measured effect sizes for eQTL VNTRs between revisions.

The quality of assemblies and proper annotations of VNTR boundaries are both crucial to the genotyping accuracy using RPGG. We identified VNTR annotation as the limiting step for performance; subsequent optimization improved the alignment quality for 98% (72,138/73,582) of the loci. The loci chosen in our first revision (Fig. R1, data points above the red dashed line at $\text{aln-}r^2=0.96$) while those in the initial submission are those to the right of the vertical green line. We note that the majority of loci not included in the newer version are because of an increased minimal alignment quality ($\text{aln-}r^2$) (Fig. R1).

Fig. R1. Comparison of $aln-r^2$ between versions. Loci annotated in the initial manuscript ($n=84,411$) were intersected with the loci annotated in our first revision ($n=73,582$), leaving a total of 74,125 and 73,582 loci in each version. The green and red dashed lines indicate the filtering threshold in our initial submission (0.8) and the first revision (0.96), respectively.

To fully address the concern of robustness, we replicated our findings in highly stratified loci and eVNTR discoveries.

While the Vst statistic was highly correlated between versions ($r^2=0.759$), the highlighted loci in our initial submission were among the outliers that had reduced Vst in the new version (*FREM1*), or were among the 9.9% not included in RPGG. However, to demonstrate replication of Vst using the same input RPGG but different genomes, we computed Vst statistics on the 698 child genomes related to the 2,504 genomes (parental) from the 1000 Genomes Project (1KGP), and compared the statistics with those computed from the 2,504 genomes. We observed a high correlation ($r^2=0.828$) between the two datasets (Fig. R2, left), indicating the robustness of our method. We observe that 89.2% (700/785) of the loci highlighted previously passed the $3\times std$ threshold ($Vst>0.107$) in the replication set (Fig. R2, right).

Fig. R2. Comparison of Vst between versions and across datasets. Left panel: the Vst from our initial submission versus the first revision. Right panel: replication of Vst on the 698 genomes related to the 1KGP samples. The 2,504 1KGP samples were retrieved from ENA project PRJEB31736. The 698 genomes were retrieved from ENA project PRJEB36890. Vst was computed over the 32,138 VNTR loci using the total k -mer dosage as proxy for length.

Finally, for the eVNTR discoveries, we included additional 2,870 loci that were genotyped in our initial manuscript (n=29,052) but not in the first revision (n=32,138). The correlation between versions is 0.999 (Fig R3), with the two previously highlighted eVNTRs *RNASET2* and *CBY1* remaining significant ($p=1.9\times 10^{-56}$ and 5.5×10^{-11} , respectively). We have included a table of the eVNTRs in the expanded set of loci in the supplementary material.

Fig R3. Correlation of effect size between versions. The previous version includes 29,052 VNTRs. The current version contains 35,008 VNTRs obtained by augmenting the 32,138 VNTRs with additional 2,870 loci that were present in the previous version but were removed in the first revision due to stringent QC threshold.

Comment 10:

For example, if the authors were to use a different set of LRS assemblies to build the pangenome graphs, it is feasible that the set of VNTRs that can be genotyped with high confidence will change a lot.

The annotation of boundaries of VNTR sequences have the greatest effect on ability to genotype using the RPGG. To test if VNTR boundaries change considerably when using a different set of assemblies, we incrementally included genomes to the RPGG according to their respective genome graph size (smallest first, beginning with 10 genomes). The boundaries of VNTR loci are stable as genomes are added (Supplementary Fig. 40, left), with 0.13% (97/73,582) of loci changed on average (cutoff=50 bp, Supplementary Fig. 40, right panel). Thus, the VNTR loci used to generate the RPGG, and the feasible set of genotypable VNTRs are highly stable with the inclusion of additional genomes.

Supplementary Figure 40. Incremental RPKG construction and change in boundary annotations. Left panel: Distribution of boundary change relative to the previous iteration of RPKG construction. Right panel: Number of loci with expansion size passing each threshold (legend) in each iteration. Δ Boundary is computed by summing the change in boundaries relative to the previous iteration and dividing the value by the number of supporting haplotypes. Boundary expansion was applied to the initial set of 84,411 loci annotated using TRF.

Comment 11:

Also, in response to comment #29 (from the response document), the authors state that "As the number of included genomes increases, the probability of the boundary in one genome not aligning with others also increases." This indicates that the process for selecting VNTR boundaries is not robust to outliers and can adversely affect the genotyping accuracy. Does this imply that VNTR pangenome graphs built using hundreds of LRS assemblies (that may be available in the future) will actually perform worse?

Response 11:

The following description "As the number of included genomes increases, the probability of the boundary in one genome not aligning with others also increases" was used in the context of our older boundary alignment approach where VNTR boundaries were annotated for each genome. To mitigate the effect of outliers on boundary annotations, we therefore updated the boundary expansion algorithm to detect and correct misaligned boundaries based on the evidence from multiple haplotypes. We expect the method to be scalable and produce RPKGs with comparable performance or better if future LRS assemblies have less errors and sample more population diversity.

Comment 12:

This problem seems similar to what has been observed in the context of read alignment to graph genomes where adding more variants to the graph can actually be detrimental to the alignment accuracy (see <https://genomebiology.biomedcentral.com/articles/10.1186/s13059-018-1595-x>).

Response 12:

We acknowledge that simply adding all variants to a graph can be detrimental, however, the FORGe method is mostly focusing on SNVs, short indels and leaves out structural variants in repeat regions that usually have a higher divergence rate. We believe that with careful quality control on assembly generation, variant annotation and graph construction, including more genomes is likely to be beneficial for genotyping in complex regions, as shown in our work and some others (Li et al. 2020; Sirén et al. 2020).

Comment 13:

Overall, this is a comprehensive piece of work on addressing an under-studied source of variation in human genomes. However, the robustness of the method (i.e. the VNTR genotyping) is still of some concern to me. Hopefully, this can be addressed in future work.

Response 13:

We thank the reviewer for recognizing the challenges in studying these complex regions, and we hope that our additional analyses show sufficient robustness to our methods.

Minor comment:

Comment 14:

I also looked at the new VNTR eQTLs reported in the paper. For the ERAP2 locus, the authors state that "This VNTR is a unique sequence in GRCh38 that is a 101 bp tandem duplication in 17/38 of the haplotypes." Does this mean that the VNTR is simply a bi-allelic variant or are there more than two alleles of the VNTR? To demonstrate that the VNTR is driving the change in gene expression, it may be feasible to test if there is SNV at the same locus that is also associated with the gene expression and if the association is weaker than the VNTR.

Response 14:

We examined the association of chr5:96896863-96896963 VNTR with *ERAP2* expression within the context of eSNP (Supplementary Fig. 22). The results indicate that the effect of VNTR is not independent of the lead eSNP at chr5:96916885 (pooled effect size=-0.010, P=0.78). The linkage disequilibrium (LD) structure indicates that the genotype of the VNTR is strongly linked to numerous nearby eSNPs (Supplementary Fig. 23). Nonetheless, we think it is worth highlighting this locus in the

main figure given that the VNTR sequence is immediately adjacent to the exon 9 of *ERAP2*, overlaps several regulatory features, and is not documented in the GTEx *cis*-eQTL catalogue as a indel variant.

We have also added the following text to the manuscript:

Although the effect is not independent of the lead eSNP (Supplementary Fig. 22-23), the variant is missing from the GTEx *cis*-eQTL catalogue and colocalizes with a regulatory hotspot with peaks of histone markers, DNase and 40 different CHIP signals.

Supplementary Figure 22. Conditional association of chr5:96896863-96896963 VNTR with *ERAP2* expression over chr5_96916885_T_C_b38. Marginal association between VNTR and expression was performed by subsetting on samples with the indicated genotype (subtitle) at the SNP site. The effect size (*b*) and P-value (*P*) for each association test was shown in each subpanel. The red dashed line indicates the regression line. HOM_REF, homozygous reference; HET, heterozygous; HOM_HET, homozygous alternative.

Supplementary Figure 23. Linkage disequilibrium (LD) between chr5:96896863-96896963 VNTR and nearby SNPs. The LD between the VNTR and each nearby SNP was computed as the r^2 between genotype values. The y-axis indicates the association P-value with *ERAP2* expression level. The location of VNTR (blue asterisk) and *ERAP2* gene (blue line) are highlighted.

Reviewer #4 (Remarks to the Author):

The authors have performed an extensive analysis to address comments from reviewers. VNTR, as one of the most divergent and under-studied categories of genomic variants, is very difficult to be called and to be benchmarked. The authors' additional analysis is solid. With the following minor issues addressed, the manuscript should be good to be published.

Comment 15:

1. The term "read sampling bias" is not clearly explained with LSB. Although the authors have changed "read sampling bias" in Figure 3a, I recommend the authors review this term across the full text and supplementary figures. Specify/replace it if needed.

Response 15:

We thank the reviewer for pointing out this confusion. We have added the following sentence to the text:

LSB measures the deviation of an observed read depth from the expected value within an interval (see Methods for formal definition).

We also replaced all "read sampling bias" with "LSB" for consistency.

Comment 16:

2. To address the reviewer comment 32, the authors added a figure showing that sampling bias is a reason for the genotyping variation in Figure 3(a). If this indicates data quality as the reason for this variation, the authors could state that.

Response 16:

Please refer to Response 18 for detailed analyses.

Comment 17:

Additionally, there is a better 30x genome of NA24385 at ENA PRJEB35491. If the authors would like to solve the quality problem of NA24385 they can redo their benchmarking with this data.

Response 17:

We thank the reviewer for the suggestion. The sequence data at ENA PRJEB35491 did improve the accuracy for NA24385 (from 0.62 to 0.75), and mildly increased the overall per-locus accuracy using RPGG and the per-genome accuracy using a read depth approach.

We have changed the following text from:

However, estimation of VNTR length from read depth has an accuracy of 0.72 (Figure 3a left). We also compared the performance for length prediction using the RPGG versus repeat-GRCh38, and observed a 58% improvement in accuracy (0.82 versus 0.52, Figure 3a left, Supplementary Fig. 11). The overall error rate, measured with mean absolute percentage error (MAPE), of all loci (n=32,138) are also significantly lower when using RPGGs (MAPE=0.19, Figure 3a right) compared with the repeat-GRCh38 (0.23, paired *t*-test $P = 1.7 \times 10^{-31}$) or reference-aligned read depth (0.21, paired *t*-test $P = 2.9 \times 10^{-31}$). Furthermore, a 61% reduction in error size is observed for the 6,238 loci poorly genotyped (MAPE > 0.4) using repeat-GRCh38 (Figure 3b, MAPE=0.233 versus 0.603).

to:

However, estimation of VNTR length from read depth has an accuracy of **0.75** (Figure 3a left). We also compared the performance for length prediction using the RPGG versus repeat-GRCh38, and observed a 58% improvement in accuracy (0.82 versus 0.52, Figure 3a left, Supplementary Fig. 11). The overall error rate, measured with mean absolute percentage error (MAPE), of all loci (n=32,138) are also significantly lower when using RPGGs (MAPE=**0.18**, Figure 3a right) compared with the repeat-GRCh38 (0.23, paired *t*-test $P = 4.2 \times 10^{-32}$) or reference-aligned read depth (**0.20**, paired *t*-test $P = 2.4 \times 10^{-33}$). Furthermore, a **62%** reduction in error size is observed for the **6,383** loci poorly genotyped (MAPE > 0.4) using repeat-GRCh38 (Figure 3b, MAPE=**0.235** versus **0.610**).

We also updated Figure 3 from:

to:

Comment 18:

3. Figure 3(a) right plot indicates variation per-locus in the three methods as well. It could be helpful for the authors to add some text and/or cases to discuss a few poorly genotyped locus and show why they do not work well in one method or all methods. The authors have already shown some factors in Supplementary Figure 28 & 29 (TR length and GC). Overall, a brief discussion or case analysis for VNTRs that do not work well with RPGG will help people to understand how difficult the VNTR calling is and to understand your method better.

Response 18:

We examined a number of different factors that could result in low prediction accuracy, including GC composition, haplotypes missing from assemblies, motifs unique to samples (novel k -mers), and finally LSB estimation error. With the exception of LSB estimation error, other characteristics were not well correlated with low accuracy of length estimation, which may be attributed to differences in characteristics of datasets between samples.

We have added the following description to the text:

Loci with low accuracy in length estimates from RPKG can be mostly explained by the estimation error in LSB due to varying data quality ($r^2=0.89$, Supplementary Fig. 12; example given in Supplementary Fig. 13), and to a slight degree by the presence of a missing haplotype (Supplementary Fig. 14), the fraction of k -mers in a locus novel to the rest of the samples (Supplementary Fig. 15), GC bias (Supplementary Fig. 16), and the difference in the VNTR GC content across samples (Supplementary Fig. 17)

Supplementary Figure 12. Correlation between the estimation error in VNTR length and in LSB. Estimation error in length was computed using absolute percentage error, i.e. $|1 - gt/est|$, where gt is the length in assembly and est is the length estimated from leave-one-out analysis. Similarly, estimation error in LSB was computed as $|1 - gt/est|$, where gt is the ground truth of the LSB for the VNTR locus (Methods) and est is the estimated LSB from the nearest neighbor (Methods). Data points were accumulated from 32,138 VNTR loci across 16 genomes.

Supplementary Figure 14. Distribution of length estimation error for loci with or without a missing haplotype. Density curves were accumulated from 32,138 VNTR loci across 16 genomes and each normalized with area 1.

Supplementary Figure 15. Correlation between length estimation error and fraction of novel *k*-mers. Fraction of novel *k*-mers for each locus in each genome was computed as the percentage of *k*-mers missing from the leave-one-out locus-RPGG. Data points were accumulated from 32,138 VNTR loci across 16 genomes.

Supplementary Figure 17. Effect of GC content change on bias and length estimation. Left panel: the correlation between GC content and LSB in VNTR regions. Middle & right panels: correlation between GC content change and length estimation error. GC content change (delta GC%) was computed from the VNTR sequence of a locus and the sequence of its nearest neighbor (same locus in another genome) in leave-one-out analysis. The analysis was restricted to HGSCV samples (HG00514, HG00733 and NA19240 trios).

Finally, we have updated the discussion to state:

Datasets combined from disparate sequencing runs with batch effects will affect dosage estimates.

Comment 19:

4. Reviewer comment 18 asks for the purity cutoff in TRF to be discussed. e.g. if a repeat is >99% pure and is short enough, it can be genotyped by GangSTR, too. Better to quantify how many of your VNTRs are like this.

Response 19:

We have added the following description to manuscript:

This filtering criterion corresponds to an empirical cutoff of 56% purity and can retain VNTRs (n=2,715, Supplementary Fig. 1) that have nested STR annotations (Supplementary Fig. 2).

Supplementary Figure 2. An example of multiple STR annotations within a VNTR. Dot plot was generated using exact matching between 9-mers along chr1:861277-862683. Annotations of four STRs (red box; chr1:861863-861874, chr1:862001-862016, chr1:862077-862088 and chr1:862133-862144) and one VNTR (blue box; chr1:861777-862183; before boundary expansion) are highlighted.

In supplementary figure 1 (also shown below for illustration), we compared the tandem repeats in the GangSTR catalogue and our VNTR set. The result (Supplementary Fig. 1, bottom right) indicates that only 8.4% (n=2715) of the VNTRs in our dataset overlap with the GangSTR catalogue.

Supplementary Figure 1. Comparison between the TR database in GangSTR and this work. **a**, Size distribution of the TRs annotated in each study. TRs with size greater than 150 bp in at least one assembly and with size greater than 50 bp in hg38 are annotated in this study. TR sizes above 1000 bp, above 50 bp and below 50 bp are not shown for this study (left), GangSTR (middle) and comparison (right), respectively. **b**, Percentage of overlapping TRs between databases. The number of overlapping loci changes across databases since multiple loci in GangSTR's database could correspond to only one locus in our database.

Text comments:

Comment 20:

1. On page 2 line 4, "single-molecule sequencing" should be "long-read sequencing" to match the abbreviation "LRS".

Response 20:

We have fixed the error accordingly.

Comment 21:

2. On page 4 line 12, "sequenced by either PacBio single long read (SLR)": I recommend replacing "SLR" with "contiguous long read" (CLR). This is the term that PacBio is officially using.

Response 21:

We have corrected the term accordingly.

Comment 22:

3. On page 6 line 6, the authors mentioned that alignment with cycles is "recently solved by GraphAligner". Actually, the sequence-graph version of ExpansionHunter (2019) also aligns reads to a local cycled graph. It could be nice to include this as well.

Response 22:

We thank the reviewer for pointing this out. We have update the text from:

While methods exist to find alignments that do not reuse cycles (Rakocevic et al. 2019), alignment with cycles is a more challenging problem recently solved by GraphAligner (Rautiainen, Mäkinen, and Marschall 2019) to map long reads to pangenome graphs and by ExpansionHunter (Dolzhenko et al. 2019) to map short reads.

to

While methods exist to find alignments that do not reuse cycles (Rakocevic et al. 2019), **others allow alignment to cyclic graphs but with high computational costs when applied to RPGG (Garrison et al. 2018) or are limited to alignment in STR regions (Dolzhenko et al. 2019).** **Efficient** alignment with cycles is a more challenging problem recently solved by GraphAligner (Rautiainen, Mäkinen, and Marschall 2019) to map long reads to pangenome graphs.

REVIEWER COMMENTS

Reviewer #1 (Remarks to the Author):

The authors have provided an excellent response to my concerns, and in doing so have substantially improved key parts of their argument.

I look forward to seeing applications of their method to the difficult parts of the pangenome.

Erik Garrison

Reviewer #3 (Remarks to the Author):

The authors' response to the comments is quite satisfactory. I have no further comments.

Reviewer #4 (Remarks to the Author):

Thank the authors for doing the extensive analysis. All my questions have been properly addressed.

Response to all reviewers:

We appreciate the reviewers' time and efforts.

Reviewer #1 (Remarks to the Author):

The authors have provided an excellent response to my concerns, and in doing so have substantially improved key parts of their argument.

I look forward to seeing applications of their method to the difficult parts of the pangenome.

Erik Garrison

Reviewer #3 (Remarks to the Author):

The authors' response to the comments is quite satisfactory. I have no further comments.

Reviewer #4 (Remarks to the Author):

Thank the authors for doing the extensive analysis. All my questions have been properly addressed.